# EFFICACY OF DATA-FREE METRICS: ROBUST AND CRITICAL EVIDENCE FROM ROBUST AND CRITICAL LAYERS

## ABSTRACT

Data-free methods for analysing and understanding the layers of neural networks have offered many metrics for quantifying notions of "strong" versus "weak" layers, with the promise of increased interpretability. We examine the robustness and predictive power of data-free metrics under randomised control conditions across a wide range of models, datasets and architectures. Contrary to some of the literature, we find strong evidence *against* the efficacy of data-free methods. We show that they are not reparametrisation-invariant even for *robust* layers, that is to say layers that can be reparametrised by re-initialisation or re-randomisation without affecting the accuracy of the model. Moreover, we also show that data-free metrics cannot be used for the arguably simpler tasks of (i) distinguishing between robust layers and critical layers, i.e. layers that cannot be reparametrised without significantly degrading the accuracy of the model, or (ii) predicting if there will be a performance difference between re-initialisation and re-randomisation. Thus, we argue that to understand neural networks, and in particular the difference between 'strong" versus "weak" layers, we must adopt mechanistic and functional approaches, contrary to the traditional Random Matrix Theory perspective.

## 1 INTRODUCTION

Understanding and interpreting deep learning models is a critical area of research, especially as the prevalence of these models increases in real-world applications. The holy grail of neural network interpretability lies in identifying computationally cheap metrics that can provide insights into the effectiveness of neural networks and their components. Data-free methods typify this endeavour by analysing the properties of the neural network parameters without regard for the data. A key example of data-free methods is Martin & Mahoney (2021), which claims to be able to predict the performance of a neural network without the requirement of test data through the use of Random Matrix Theory to analyse the layer weight matrices. In contrast, data-dependent layer analysis via mechanistic interpretability or functional analysis attempts to quantify how inputs interact at specific layers and use comparative analysis to understand the interaction between model parameters and data (Olah et al., 2020; Klabunde et al., 2025; Nanda et al., 2023).

Zhang et al. (2022) identified an interesting and unexpected phenomenon in neural network layers: some layers within a network are robust, while others are critical. A critical layer is a layer that cannot be re-initialisatised or re-randomised without dramatically affecting the performance of the network. In contrast, a robust layer can be either re-initialisatised or re-randomised without any noticeable effect on performance. Re-initialisation sets the layer back to its initial parameters before training, whilst re-randomisation sets the parameters of the layer to random values by re-sampling from the same distribution used for initialisation. It was observed that in some cases, re-initialisation and re-randomisation can result in significant performance differences for a given layer, with re-initialisation maintaining performance but re-randomisation significantly degrading it (Zhang et al., 2022). In other cases, re-initialisation and re-randomisation of a layer lead to a negligible difference in performance.

We follow in the footsteps of Dinh et al. (2017) which studied another type of metrics, namely metrics for minima flatness and how they (fail to) relate to generalisation. In particular, the authors make the following point: "*Since we are interested in finding a prediction function in a given family*

*of functions, no reparametrisation of this family should influence generalisation of any of these functions*". In the same spirit, the robustness phenomenon suggests that certain reparametrisations of this family – re-initialisations and re-randomisations of robust layers – should not influence the functional behaviour of any of these functions (as quantified by accuracy). We believe that this provides a strong basis for asking:

- Are data-free metrics reparameterisation-invariant, particularly under re-initialisations or re-randomisations of robust layers?
- Can data-free metrics distinguish robust layers from critical layers?
- Can data-free metrics predict performance difference between re-initialisation and re-randomisation of a layer?

In contrast to Dinh et al. (2017), our approach is purely empirical and our experiments show that across data modalities, architectures and datasets:

- Data-free metrics are not invariant under reparametrisation, even when we restrict our attention to re-initialisations and re-randomisations of robust layers,
- Current data-free analyses have no predictive capacity to identify robust and critical layers in small scale experiments, nor to predict performance difference between re-initialisation and re-randomisation,
- For Large Language Models (LLMs) and ImageNet vision models norm-based metrics have no predictive capacity over the change in performance under re-randomisation.

We hope that our work can push the field forward in two core ways.

- To provide a strong test ground for the development of data-free metrics that are able to disambiguate between robust and critical layers. Similar to how the work of Dinh et al. (2017) led to the development of sharpness metrics Relative Flatness (Petzka et al., 2021) and Fisher Rao norm (Liang et al., 2019) that are reparameterisation invariant and now considered state of the art sharpness metrics.
- Offering concrete evidence for the prioritisation of data-based metrics, until the appropriate data-free metrics are discovered, such as those that we examine in the paper to improve efforts in interpretability and compression.

## 2 BACKGROUND

Data-free methods of interpretability aim to understand the inner workings of neural networks by studying the properties of the network parameters. Data-free approaches often focus on the matrix norm properties of layer weight matrices to understand learning or improve the performance of neural networks (Yunis et al., 2024; Martin et al., 2021; Sanyal et al., 2020; Feng et al., 2022; Salimans & Kingma, 2016; Bartlett et al., 2017) (Wei et al., 2022; Qing et al., 2024; Thamm et al., 2024; He et al., 2025). However, Zhang et al. (2022) showed in their work that matrix norms, such as the Frobenius norm, are too coarse to understand the generalisation properties of neural networks. Martin & Mahoney (2021) use Random Matrix Theory (RMT) to analyse the weight matrices (excluding biases) of neural network layers through training to create a theory of heavy-tailed self-regularisation. With this theory, they construct a set of predominately power-norm metrics related to generalisation that is applied after training to assess layer performance: alpha ($\alpha$), alpha-weighted ($\hat{\alpha}$), log alpha norm, and MP soft rank Martin & Mahoney (2021). In this work, they identified a value of $\alpha$ between 2 and 6 as a property of a good, well-trained layer, whereas $\alpha > 6$ indicates that a layer is underfitted and $\alpha < 2$ indicates that it is overfitted. Martin et al., Martin et al. (2021) showed a correlation between these metrics and the generalisation performance of pre-trained models in language and computer vision tasks.

The theory of heavy-tailed self-regularisation (Martin & Mahoney, 2021) has been used to provide justification for layer-wise pruning ratios in large language models (Lu et al., 2024), additionally it has been used to explain and understand stages of the grokking phenomenon (Power et al., 2022) namely, pre-grokking, grokking and anti-grokking (Prakash & Martin, 2025). The promise of understanding

both redundancy in neural networks and transitions from memorisation to generalisation means that RMT promises a lot with respect to improving the interpretability of deep neural networks, a topic of great importance given their increasing adoption across a range of different data domains.

Since Zhang et al. (2022) established that norm-based methods are ineffective, our work will focus on alpha, alpha-weighted, log alpha norm, MP Soft Rank and Generalized von-Neumann Matrix Entropy (Martin & Mahoney, 2021) as well as Frobenius Norm, Spectral Norm and Stable Rank (Rudelson & Vershynin, 2007) within the critical and robust layer phenomenon to establish whether these metrics are invariant under re-initialisation and re-randomisation of robust layers and if they can disambiguate between (i) robust and critical layers, and (ii) the performance difference between re-initialisation and re-randomisation of a layer.

## 3 EXPERIMENTAL SETUP

Zhang et al. (2022), showed the robust and critical layer phenomenon across a range of trained architectures, MLPs, VGGs (Simonyan & Zisserman, 2015), ResNets (He et al., 2016), Transformers (Vaswani et al., 2017), Vision Transformers (Dosovitskiy et al., 2021a), MLPMixers (Tolstikhin et al., 2021) across datasets MNIST (LeCun et al., 1998), CIFAR10 (Krizhevsky & Hinton, 2009), ImageNet (Deng et al., 2009) and LM1B(Chelba et al., 2014). We first choose the simplest model (ReLU FCN 5x512), Figure 1, and dataset (MNIST (LeCun et al., 1998)) identified by Zhang et al. (2022) that demonstrates this phenomenon to systematically explore the behaviour of data-free metrics under layer re-initialisations and re-randomisations of a large number of trained models.

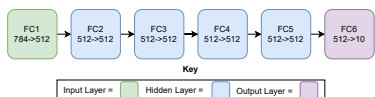

Figure 1: ReLU FCN 5x512 Model Architecture.

An added benefit of the ReLU FCN 5x512 model is that it also offers a clear performance contrast between re-initialisation and re-randomisation. Zhang et al. (2022) showed that residual blocks are robust to re-randomisation and attributed this to the residual layer potentially playing a lesser role in the network and thus having smaller activations than the skip connection. To analyse how effective the data-free metrics are at disambiguating between re-initialisation and re-randomisation, we analyse the correlation between data-free metrics and test accuracy using the Spearman correlation coefficient $\rho$, the root mean square error (RMSE) of the linear regression and Kendall's tau measure (K-$\tau$). Where $\rho$ and K-$\tau$ score of -1 indicates a very strong negative correlation, 0 indicates no correlation, and 1 indicates a very strong positive correlation. We use RMSE and Kendall's Tau measure for this study as they are two of the correlation metrics used in Martin et al. (2021) to highlight the predictive capacity of the data-free metrics.

We trained 100 ReLU FCN 5x512 models, creating 100 initialisations and 100 trained models, to obtain a representative sample of possible initialisations and trained models. The model weights and biases are initialised and re-randomised from the same distribution $\mathcal{U}(-\sqrt{k}, \sqrt{k})$ where $k$ is $\frac{1}{\text{in\_features}}$, e.g. FC1 has $k = \frac{1}{784}$. We record the data-free metric properties of these trained models' layers when they undergo re-initialisation and re-randomisation and observe whether these metrics:

(a) are invariant, particularly for robust layers,

(b) can distinguish critical and robust layers,

(c) can predict the performance difference between re-initialisation and re-randomisation.

This exploration also demonstrates the overall predictive capacity of data-free metrics after training.

**Data-Free Metrics.** Power Norm based data-free methods analyse a layer weight matrix, $W$, excluding the bias. A variety of data-free metrics have been developed in the literature to quantify the importance of a layer, we focus on the following metric (Martin & Mahoney, 2021):

- **Alpha ($\alpha$):** The fitted power law exponent, $\alpha$, for the empirical spectral density of the correlation matrix $X = W^T W$, such that $p_{emp}(\lambda) \sim \lambda^{-\alpha}$, with $\lambda$ the eigenvalues of $X$.

In section 4.2 we additionally explore the following metrics:

- **Alpha Weighted ($\hat{\alpha}$):** $\alpha \log(\lambda_{max})$, where $\lambda_{max}$ is the max eigenvalue from $X$ Martin & Mahoney (2021).

- **Log Alpha Norm:** $\log(||X||_\alpha^\alpha)$, where $||X||_\alpha^\alpha = \sum_i^M \lambda_i^\alpha$, where $M$ is the rank of $W$ Martin & Mahoney (2021).

- **MP Soft Rank:** is the ratio between the bulk edge of the $p_{emp}(\lambda)$, $\lambda^+$, and the max eigenvalue, $\lambda_{max}$, $\frac{\lambda^+}{\lambda_{max}}$ Martin & Mahoney (2021).

- **Spectral Norm**: The max singular value of $W$ denoted as $||W||_\infty$.

- **Stable Rank:** The ratio of the squared Frobinues Norm and the squared Spectral Norm, denoted as $\frac{||W||_F^2}{||W||_2^2}$ Rudelson & Vershynin (2007).

- **Generalized von-Neumann Matrix Entropy:** $\frac{-1}{log(M)} \sum_i p_i \log p_i$, where $M$ is the rank of matrix $W$ and $p_i$ is $\frac{\sigma_i^2}{\sum_i(\sigma_i^2)}$ where $\sigma$ is the singular values of $W$ Martin & Mahoney (2021).

The metrics are collected using the weightwatcher[1] tool.

We extend our analysis of questions (a) and (b) to large-scale pre-trained computer vision models (ResNet34 [2] (Wightman et al., 2021; Wightman, 2019; He et al., 2016) and ViT[3] (Steiner et al., 2021; Dosovitskiy et al., 2021b; Wightman, 2019), both trained on ImageNet(Russakovsky et al., 2015)) and pre-trained large language models (GPT2[4] and GPT2-Large[5] (Radford et al., 2019) evaluated on WikiText103 (Merity et al., 2016)), in Section 5.

## 4 RESULTS AND DISCUSSION FOR SMALL SCALE REPARAMETRISATIONS

For clarity and succinctness, we primarily present our results for alpha ($\alpha$) of Martin & Mahoney (2021) in the body of the paper, however in Section 4.2 we show that these result generalise to additional metrics. In Appendix Section A we present the analysis of the Frobenius Norm.

### 4.1 ANALYSIS OF ALPHA UNDER RE-INITIALISATION AND RE-RANDOMISATION

Table 1: Alpha ($\alpha$) of the layers in ReLU FCN 5x512 and test accuracy of the model. Mean and $\pm$ 1 SEM (Belia et al., 2005) (Standard Error from the Mean) derived from 100 trained models on MNIST.

| Metric | Layer | | | | | | Test Accuracy |
|---|---|---|---|---|---|---|---|
| | FC1 | FC2 | FC3 | FC4 | FC5 | FC6 | |
| Alpha ($\alpha$) | $4.82 \pm 0.025$ | $4.205 \pm 0.039$ | $4.126 \pm 0.038$ | $4.135 \pm 0.035$ | $4.193 \pm 0.034$ | $3.793 \pm 0.805$ | $96.822 \pm 0.057$ |

**Quality of training vs alpha.** Data-free metrics such as $\alpha$ aim to identify well-trained layers, with a well-trained layer having a value of $\alpha$ between 2 and 6 Martin & Mahoney (2021). This seems to be borne out by training 100 ReLU FCN 5x512 models to good values of test accuracy, achieving $\alpha \in [2, 6]$ for every layer (see Table 1). Whilst well-trained may imply $\alpha \in [2, 6]$ (but more on this below), a simple experiment shows that the converse does not hold. We plot the empirical distribution of $\alpha$ for a 512x512 fully connected layer in Figure 2 (left), sampled from 10,000 potential

---

[1]https://weightwatcher.ai

[2]https://huggingface.co/timm/resnet34.tv_in1k

[3]https://huggingface.co/timm/vit_base_patch16_224.augreg_in1k

[4]https://huggingface.co/openai-community/gpt2

[5]https://huggingface.co/openai-community/gpt2-large

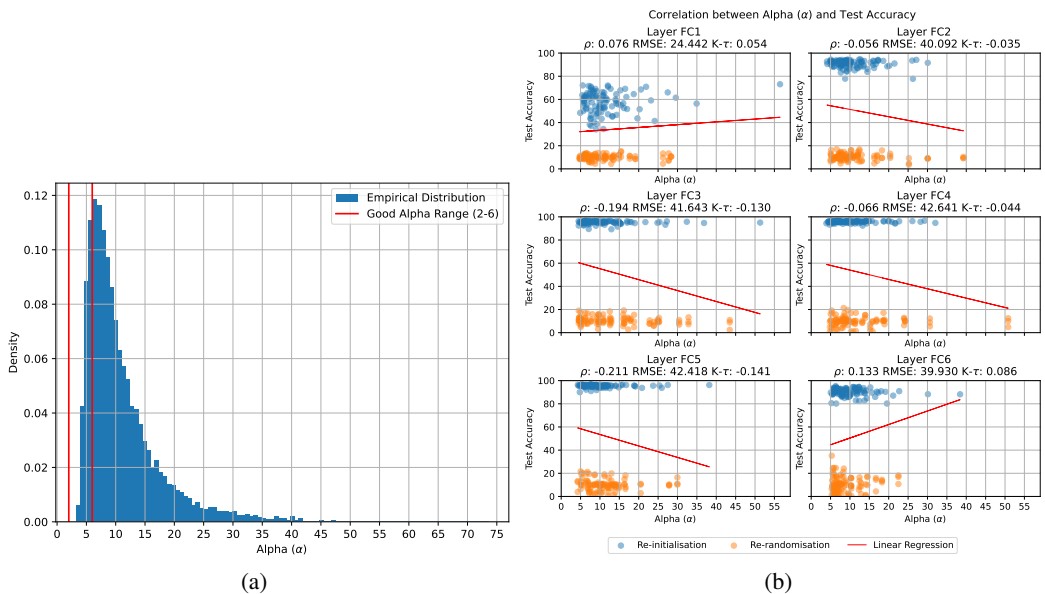

(a)  (b)

Figure 2: Empirical distribution of $\alpha$ values on a 512x512 fully connected layer, sampled from 10,000 initialisations (**Left**). Layer re-initialisation (**blue**) and re-randomisation (**orange**) test accuracy vs $\alpha$, $\rho$ is the Spearman correlation coefficient, $RMSE$ is the root mean square error of the linear regression (red line), and K-$\tau$ is the Kendall's tau measure, all with respect to the relationship between test accuracy and $\alpha$ values (**Right**).

initialisations. The resulting distribution shows that an initialised, *untrained* layer of this network, can fall, with a small but non-negligeable probability, within the optimal $\alpha$ value range of 2 and 6.

Next, we perform independent re-initialisations (**blue** in Figure 2) and re-randomisations (**orange**) of each layer for 100 trained ReLU FCN 5x512 networks and record the impact on the networks' test accuracy and alpha values. Figure 2 shows that a non-negligeable proportion of networks whose performance is severely degraded by re-randomisation maintain a good value of $\alpha$ around 5. Conversely, many networks maintain good performance after re-initialisations (particularly of layers FC3-FC5) but with $\alpha$ values significantly outside of $[2, 6]$. Good models can have bad alphas, bad models can have good alphas.

**Reparametrisation-invariance of alpha.** As is clear from 2, whilst all values of $\alpha$ start in the "good" range $[2, 6]$, reparametrisation in the form of re-initialisation or re-randomisation scatters these values over a very wide range (typically deep into "underfitted" territory)), irrespective of the change in accuracy incurred by this reparametrisation. Alpha is not invariant under these reparametrisations, even when accuracy is (i.e. on robust layers).

**The robust vs critical phenomenon and alpha** Figure 2 shows the stark contrast in how layers respond to re-initialisation and re-randomisation. For example, we only see a large drop in test accuracy when applying re-initialisation to the Layer FC1, re-initialising other layers leaves performance almost unchanged. From this perspective, FC1 is a critical layer whilst FC2-FC6 are robust (to re-initialisation). In this experiment $\alpha$ cannot distinguish between these behaviours, the range of values taken by $\alpha$ shows no noticeable difference between critical and robust layers.

**The re-initialisation vs re-randomisation phenomenon and alpha.** We observe different results when re-randomisation is applied. We find that re-randomising any layer degrades accuracy to circa random accuracy on the test set, in other words none of the layers are robust to re-randomisation. Surprisingly, this is not reflected in the corresponding $\alpha$ values of these two conditions, as the distribution of $\alpha$ values is relatively similar for each layer and each condition. If $\alpha$ had predictive power we would expect to observe a negative correlation between re-initialisation (corresponding

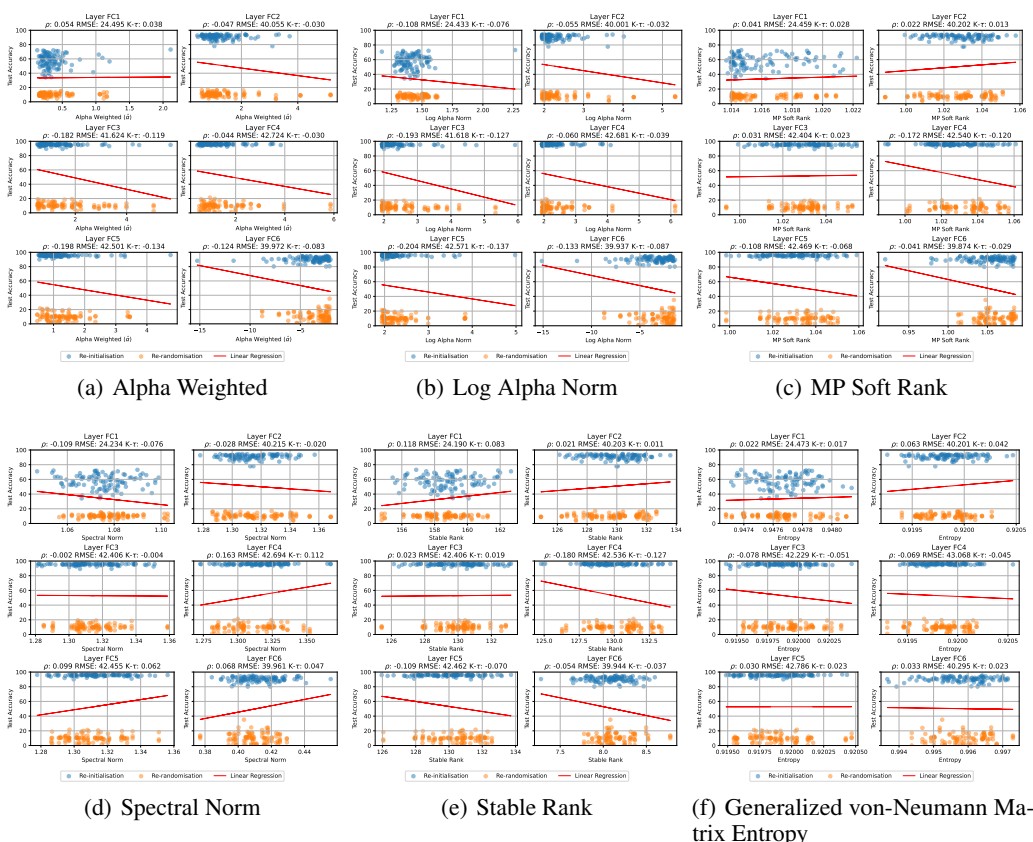

Figure 3: Layer re-initialisation (**blue**) and re-randomisation (**orange**) test accuracy vs the respective metric. $\rho$ is the Spearman correlation coefficient, $RSME$ is the root mean square error of the linear regression (red line), and K-$\tau$ is the Kendall's tau measure, all with respect to the relationship between test accuracy and the metric.

to "good" alphas) to re-randomisation (corresponding to "bad" alphas). However, there is almost no difference between the $\alpha$ values of re-initialisation and re-randomisation, with a mean Spearman correlation coefficient and Kendall's tau measure across layers of -0.053 and -0.035, respectively. Thus $\alpha$ does not distinguish between these behaviours either. These findings extend to other data-free metrics in Appendix Section A and Section 4.2. Moreover, in Appendix D, we show that $\alpha$ is not causally related to generalisation through two experiments. The first observes how random performance can emerge even when all layers are within the "good" $\alpha$ range ($2 \leq \alpha \leq 6$ and the second shows that training models within and outside of the "good" $\alpha$ range can result in models with equivalent test accuracy.

## 4.2 GENERALISATION OF REPARAMETRISATION INVARIANCE TO DATA FREE METRICS

We extend our analysis to other data-free metrics that have been defined in literature, we argue that given the lack of predictive power of the $\alpha$ metric in the previous section, it is important to verify across other representative measures that data-free metrics broadly cannot disambiguate between critical and robust layers, between re-initialisation and re-randomisation. When we conduct a similar analysis across six other metrics, Alpha Weighted, Log Alpha Norm, MP Soft Rank, Spectral Norm, Stable Rank and Generalized von-Neumann Matrix Entropy we find, as shown in Figure 3, that we can draw exactly the same conclusion as with $\alpha$.

### 4.3 Comparing To Data-Based Metrics

To compare data-free and data-based metrics we explore if two widely used data-based metrics can distinguish the output of a re-initialised layer from that of a re-randomised layer.

The outputs of the original and modified layers are represented as $\mathcal{O}_l$ and $\mathcal{M}_l$, respectively. The model layer outputs are collected by passing through the test dataset, $\mathcal{D}_{test}$. The metrics explored are defined as follows (Linear CKA (Kornblith et al., 2019) is explored in Appendix C):

- **Activation Disagreement:** The mean percentage of times that the neurons from $\mathcal{O}_l$ and $\mathcal{M}_l$ fail to agree to activate on $\mathcal{D}_{test}$.
- **Jensen–Shannon (JS) Divergence:** The mean weighted average of the Kullback–Leibler (KL) Divergence of the softmax( $\mathcal{O}_l$ on $\mathcal{D}_{test}$) compared to the softmax( $\mathcal{M}_l$ on $\mathcal{D}_{test}$) (Lin, 1991).

When considering the results of the activation disagreement and JS divergence between the original model layer and each model with re-initialised and re-randomised layers in Figure 4, it becomes clear that there is a stark difference in similarity between the re-initialised and re-randomised layers compared to their original layer. The figure demonstrates that the difference between re-initialisation and re-randomisation can be explained by the amount the layer disagrees on activation compared to the original non-modified layer. It is evident that when there is less disagreement (re-initialisation), the model can maintain accuracy, while a lot of disagreement (re-randomisation) leads to a drop in accuracy. This observation is further supported by a mean Spearman correlation coefficient and Kendall's tau measure across layers of -0.776 and -0.5375, respectively for Activation Disagreement, indicating a negative relationship between activation disagreement and test accuracy.

We have produced these results to show that there exist metrics that can disambiguate between re-initialisation and re-randomisation behaviours which we believe deserve increased focus over data-free metrics (whether they can distinguish critical from robust is, in this instance, less clear).

In Appendix B, we provide an analysis of data-based metrics via the Lottery Ticket Hypothesis (Frankle & Carbin, 2018), which shows that the ability to disambiguate robust and critical layers is related to considerations of subnetwork pathways, an inherent consideration of the data-based metrics we survey.

## 5 Results and Discussion For Large Scale Reparametrisations

We further explore how predictive the $\alpha$ metric can be when predicting performance of layers in large scale models that have been trained for high-complexity tasks using the re-randomisation setup. Overall we find, consistent with our small scale experiments, that the $\alpha$ metric has no predictive capacity under this condition, therefore questioning its utility in predicting the performance of layer criticality and "trainedness". For these results we employ pre-trained open source models, therefore, we do not have access to the starting initialisation and, as a result, we only focus on the case of re-randomisation to test the predictive capacity of the $\alpha$ metric at this scale. We show the relationship reported in this section is consistent with other data-free metrics in this experimental set up in Appendix Sections A.2, A.3, A.4 and A.5.

### 5.1 ImageNet Scale

We explore the relationship of a layer's $\alpha$ value with the corresponding performance of the model on ImageNet with pre-trained ResNet34 and ViT models. The respective performance and model details are shown in Table 2; from this table it can be observed that these models are well trained for the task of ImageNet. In line with the Random Matrix Theory perspective, the competitive performance of these models is aligned with a mean $\alpha$ value across layers within the optimal range, representing two well-fitted networks.

We employ both the pre-trained ResNet34 and ViT as base models for our re-randomisation experiment. To conduct this experiment we randomise a layer independently and then record the corresponding test accuracy and $\alpha$ value post re-randomisation. Given that different implicit biases respond uniquely to re-randomisation we believe that our selection of models is representative for

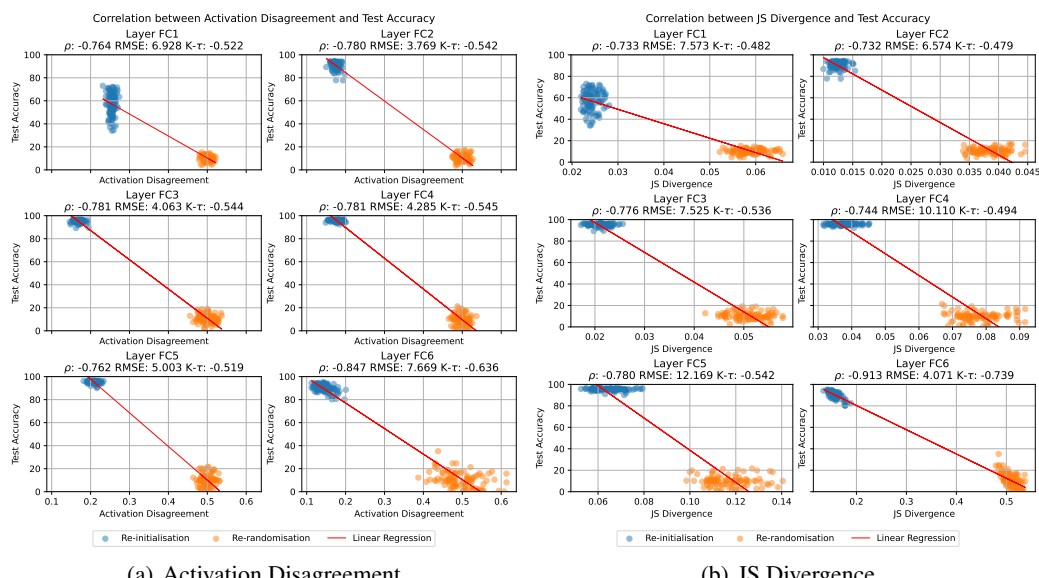

(a) Activation Disagreement        (b) JS Divergence

Figure 4: Layer re-initialisation (**blue**) and re-randomisation (**orange**) test accuracy vs the respective metric. $\rho$ is the Spearman correlation coefficient, $RSME$ is the root mean square error of the linear regression (red line), and K-$\tau$ is the Kendall's tau measure, all with respect to the relationship between test accuracy and the metric.

the core architectural differences, allowing a robust analysis of the impacts of re-randomisation and $\alpha$ metrics at scale. We repeat our re-randomisation process 10 times per layer to capture stochastic variation in randomisations. In both ResNet34 and ViT, Figure 5, we find no relationship between $\alpha$ and test accuracy, as found in the small scale experiment. The mean Spearman correlation coefficients are 0.006 and -0.051, and Kendall's tau values are 0.005 and -0.035, respectively.

Table 2: ResNet34 and ViT performance on ImageNet and mean layer Alpha ($\alpha$) value $\pm$ 1 SEM (Belia et al., 2005) (standard error from the mean).

| Model | Number of Parameters | Loss | Accuracy | Mean Layer Alpha Value |
|---|---|---|---|---|
| ResNet34 | 21.8M | 1.071 | 73.302 | $3.380 \pm 0.195$ |
| ViT | 86.6M | 0.825 | 78.852 | $4.711 \pm 0.271$ |

As previously reported in the small-scale experiments, layers in vision models can be re-randomised to $\alpha$ values outside of the desired range but continue to retain accuracy. While in the case of the ResNet34 we see that the model is not so robust to re-randomisation, having a lower accuracy than the baseline, there is a large proportion of layers with optimal $\alpha$ values that result in a model with essentially random accuracy. The ViT is more robust to layer re-randomisation and show the inverse case where many layers are significantly outside of the optimal $\alpha$ range but display a high preservation of the baseline accuracy. As a result, for large scale vision models these results confirm that $\alpha$ is not invariant under reparametrisation by re-randomisation, even on robust layers, and has very limited predictive capacity over post re-randomisation performance.

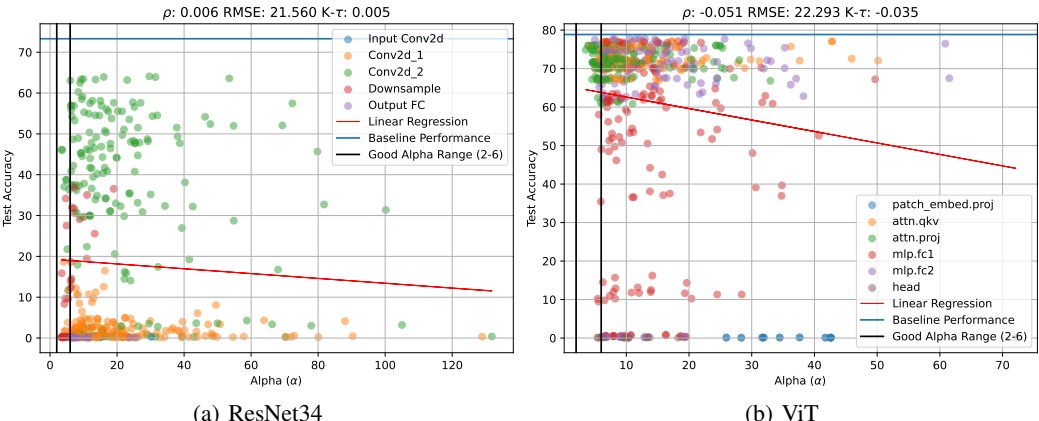

(a) ResNet34           (b) ViT

Figure 5: ResNet34 and ViT layer re-randomisation Test Accuracy vs Alpha ($\alpha$) on ImageNet. $\rho$ is the Spearman correlation coefficient, $RMSE$ is the root mean square error of the linear regression (red line), and K-$\tau$ is the Kendall's tau measure, all with respect to the relationship between the performance and $\alpha$ values.

## 5.2 Large Language Models

Given our negative results for the predictive capacity of the $\alpha$ metric in the vision domain under re-randomisation, it is important to verify the generality of our findings under a different modality. Therefore, we have extended the exploration of the relationship of a layers $\alpha$ value and the corresponding performance of the model to language models on WikiText103 with pre-trained GPT2 and GPT2-Large, the respective performance and model details is shown in Table 3. These models represent an increased scale of our experiments as they have hundreds of millions of parameters. Both models have competitive performance with low loss and perplexity on the WikiText-103 datasets and, in-line with Random Matrix Theory have mean layer $\alpha$ values that are within the optimal range.

Table 3: GPT and GPT-Large Test performance on WikiText-103 and mean layer Alpha ($\alpha$) value $\pm$ 1 SEM (Belia et al., 2005) (standard error from the mean).

| Model | Number of Parameters | Loss | Perplexity | Mean Layer Alpha Value |
|---|---|---|---|---|
| GPT | 127M | 3.399 | 29.941 | $3.865 \pm 0.136$ |
| GPT-Large | 774M | 2.967 | 19.436 | $4.013 \pm 0.091$ |

We use the pre-trained models as the base model and then we randomise a layer and record its $\alpha$ value and the corresponding test perplexity and loss of the model. We do this 10 times for each layer to obtain a representative set of random layers. For both GPT2 and GPT2-Large, Figure 6 we observe that their is no relationship between the performance of the model and a layer's $\alpha$ value, with a mean Spearman correlation coefficient of 0.028, 0.116 and Kendall's tau measure of 0.018 and 0.078 for GPT2 and GPT2-Large receptively. We also find that randomising layers in the GPT2 and GPT2-Large model often has a negligible affect on the performance of the model irrespective of the $\alpha$ value associated with the layer. Furthermore, when re-randomising a layer and achieving negligible performance degradation we can observe that the bulk of these layers are far outside the optimal range provided by the $\alpha$ metric. Again, we see that $\alpha$ is not invariant under reparametrisation by re-randomisation, even on robust layers, and has no predictive power over performance.

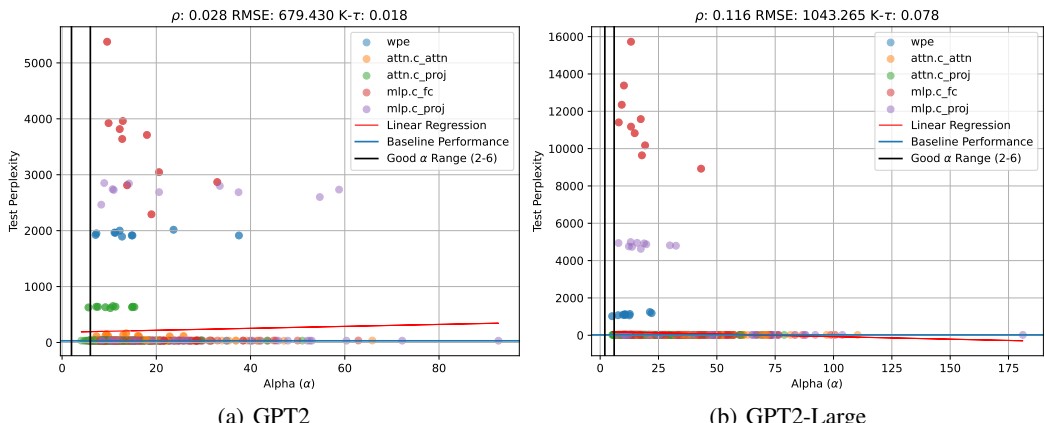

(a) GPT2  (b) GPT2-Large

Figure 6: GPT2 amd GPT2-Large layer re-randomisation Test Perplexity vs Alpha ($\alpha$) on Wiki-Text103. $\rho$ is the Spearman correlation coefficient, $RMSE$ is the root mean square error of the linear regression (red line), and K-$\tau$ is the Kendall's tau measure, all with respect to the relationship between the performance and $\alpha$ values.

## 6 CONCLUSION

In this work we concretely showed that data-free metrics cannot explain the robust vs critical layer phenomenon, nor the re-initialisation vs re-randomisation phenomenon. Based on experiments covering a wide range of these metrics and a wide range of models, we argue that they have little to no predictive capacity over these important performance-related layer properties. We highlighted how data-free metrics can be described as non-reparameterisation invariant even for robust layers for which accuracy is (to some degree) reparameterisation invariant. Our results scaling from MNIST to ImageNet and Large Language Models confirm the generality of the robustness of our findings. Finally we argue that the correlations of the alpha (and other data-free metrics) fail because they do not capture the nuanced interplay between both the data passed through the model and the weights that process it.

Concretely our recommendations are as follows:

- Avoid the use of data-free that only have spurious relations with model performance and are not a necessary prerequisite of generalisation.
- Ensure that novel data-free metrics are able to meaningfully capture performance differences offered by re-initialisation and re-randomisation.
- In lieu of such data-free metrics, to pursue understanding neural networks with metrics that consider the crucial interplay between weights and data.
- Prioritise data-based approaches for interpretability due to their capability of capturing the meaningful differences between re-initialisation and re-randomisation.

As a consequence, our results advocate for a reappraisal of the way that we approach interpretability of neural networks. Instead of using metrics which lack predictive capacity, we argue that there is a requirement for an in-depth exploration of data-free methods that can suitably disambiguate between the robust and critical layer phenomena. Or rather, a focus on methods that consider more than just weight distributions of models, which we show can be arbitrarily reparametrised without impacting the performance, and instead seek metrics which further understand the more nuanced interplay between between weights and data.

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

## A FURTHER ANALYSIS ON DATA-FREE METRICS

In this section we extend our analysis to the following data free metrics in our existing experimental setup. $W$ represents the weight matrix of the layer and $X$ is for the empirical spectral density of the correlation matrix, $X = W^T W$, such that $p_{emp}(\lambda) \sim \lambda^{-\alpha}$, where $\lambda$ are the eigenvalues of $X$.

- **Alpha Weighted** ($\hat{\alpha}$): $\alpha \log(\lambda_{max})$, where $\lambda_{max}$ is the max eigenvalue from $X$ Martin & Mahoney (2021).

- **Log Alpha Norm:** $\log(||X||_\alpha^\alpha)$, where $||X||_\alpha^\alpha = \sum_i^M \lambda_i^\alpha$, where $M$ is the rank of $W$ Martin & Mahoney (2021).

- **MP Soft Rank:** is the ratio between the bulk edge of the $p_{emp}(\lambda)$, $\lambda^+$, and the max eigenvalue, $\lambda_{max}$, $\frac{\lambda^+}{\lambda_{max}}$ Martin & Mahoney (2021).

- **Frobenius Norm**: The sum of the singular values of $W$ denoted as $||W||_F$.

- **Spectral Norm**: The max singular value of $W$ denoted as $||W||_\infty$.

- **Stable Rank:** The ratio of the squared Frobinues Norm and the squared Spectral Norm, denoted as $\frac{||W||_F^2}{||W||_2^2}$ Rudelson & Vershynin (2007).

- **Generalized von-Neumann Matrix Entropy:** $\frac{-1}{log(M)} \sum_i p_i \log p_i$, where $M$ is the rank of matrix $W$ and $p_i$ is $\frac{\sigma_i^2}{\sum_i (\sigma_i^2)}$ where $\sigma$ is the singular values of $W$ Martin & Mahoney (2021).

Each metric can be found below with the appropriate subsection that corresponds to our analysis of these data-free metrics.

The correlation between the Frobenius Norm and the associated test accuracy when layers undergo re-initialisation (blue) or re-randomisation is shown in Figure 7. The Frobenius Norm observes approximately zero correlation between the metric values and the test accuracy, highlighting the same findings as in the body of the paper.

## A.1 FROBENIUS NORM

Norm-based metrics were originally shown to be too coarse a metric to measure the generalisability of the neural networks in Zhang et al. (2022). Figure 7 strengthens these findings, highlighting that there is essentially no correlation between the Frobenius Norm of a layer and the test accuracy of a model.

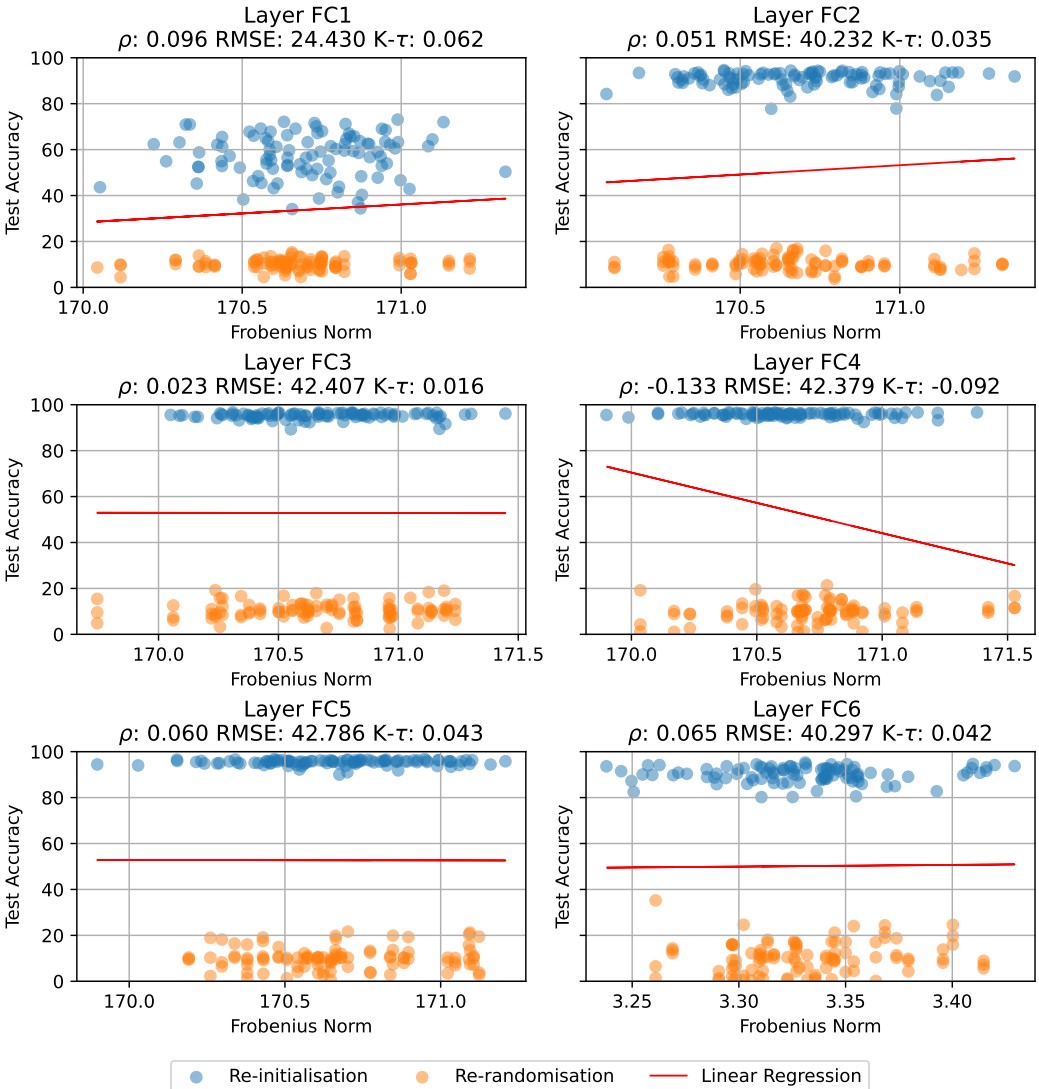

Figure 7: Layer re-initialisation (**blue**) and re-randomisation (**orange**) test accuracy vs Frobenius Norm. $\rho$ is the Spearman correlation coefficient, $RSME$ is the root mean square error of the linear regression (red line), and K-$\tau$ is the Kendall's tau measure, all with respect to the relationship between test accuracy and alpha values.

## A.2 RESNET34 ON IMAGENET

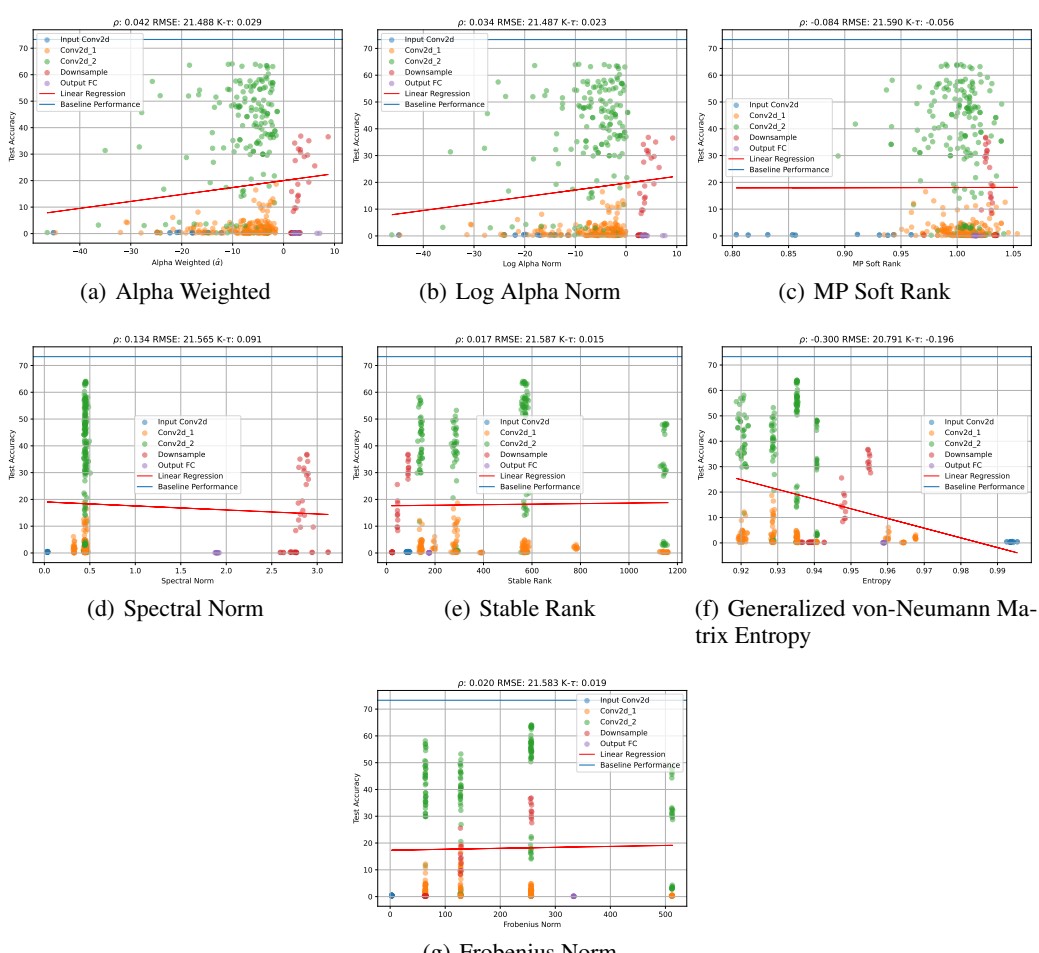

Figure 8: ResNet 34: Layer re-randomisation test accuracy vs the respective metric. $\rho$ is the Spearman correlation coefficient, $RSME$ is the root mean square error of the linear regression (red line), and K-$\tau$ is the Kendall's tau measure, all with respect to the relationship between test accuracy and the metric.

In Figure 8, we observe the same trend as found with $\alpha$ in the main body of the paper. While some correlations are higher than circa 0, i.e. (f) we can clearly observe that there is strong overlap between the values and the accuracy and that is induced by the different layer types explored.

## A.3 ViT on ImageNet

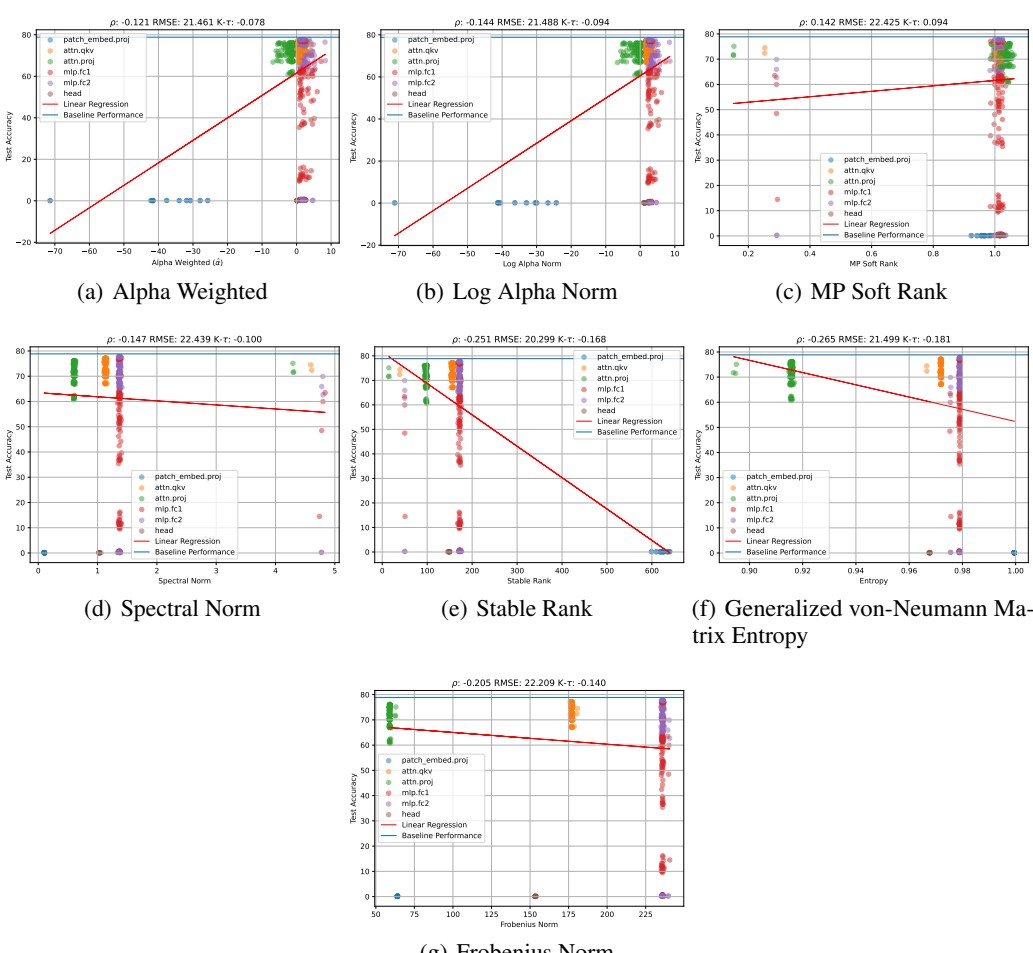

Figure 9: ViT: Layer re-randomisation test accuracy vs the respective metric. $\rho$ is the Spearman correlation coefficient, $RSME$ is the root mean square error of the linear regression (red line), and K-$\tau$ is the Kendall's tau measure, all with respect to the relationship between test accuracy and the metric.

In Figure 9, we observe the same trend as found with $\alpha$ in the main body of the paper. While some correlations are higher than circa 0, i.e. (f) we can clearly observe that there is strong overlap between the values and the accuracy and that is induced by the different layer types explored.

## A.4 GPT2 ON WIKITEXT103

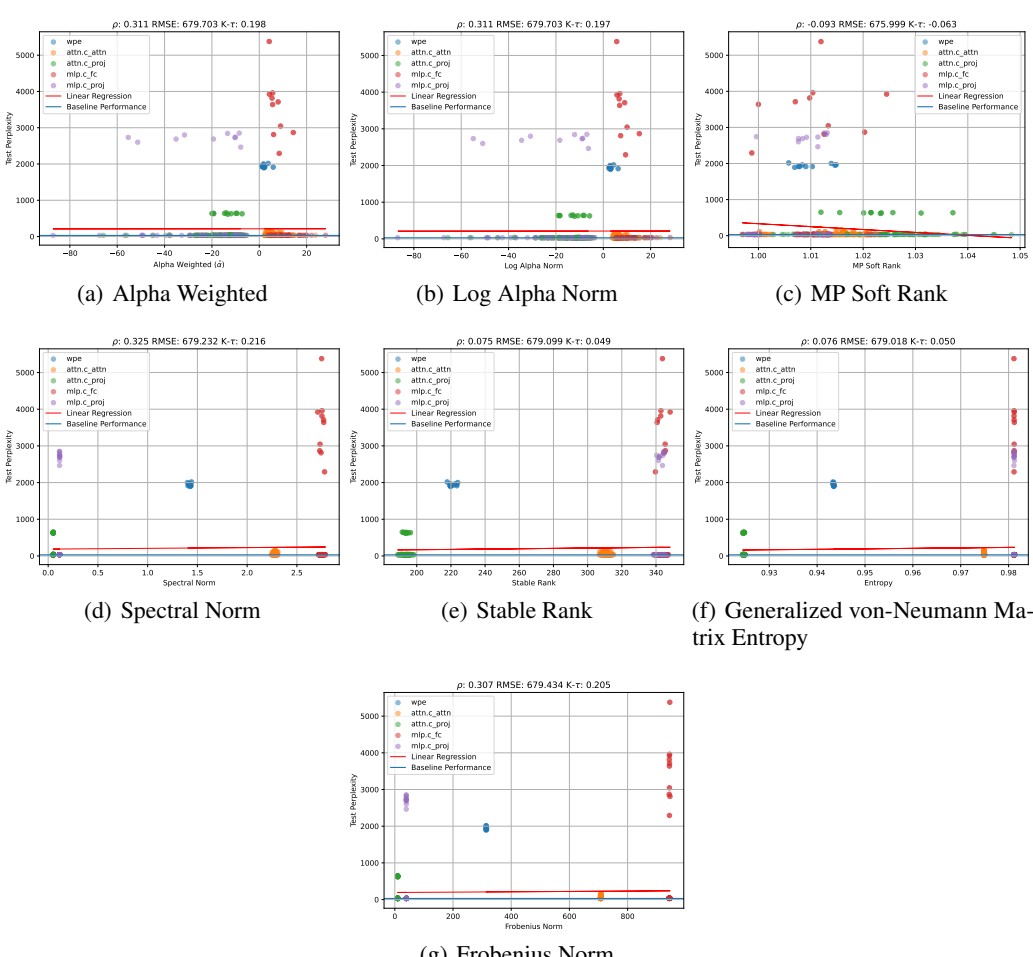

(a) Alpha Weighted

(b) Log Alpha Norm

(c) MP Soft Rank

(d) Spectral Norm

(e) Stable Rank

(f) Generalized von-Neumann Matrix Entropy

(g) Frobenius Norm

Figure 10: GPT2: Layer re-randomisation test accuracy vs the respective metric. $\rho$ is the Spearman correlation coefficient, $RSME$ is the root mean square error of the linear regression (red line), and K-$\tau$ is the Kendall's tau measure, all with respect to the relationship between test accuracy and the metric.

In Figure 10, we observe the same trend as found with $\alpha$ in the main body of the paper, across all metrics explored.

## A.5 GPT2-LARGE ON WIKITEXT103

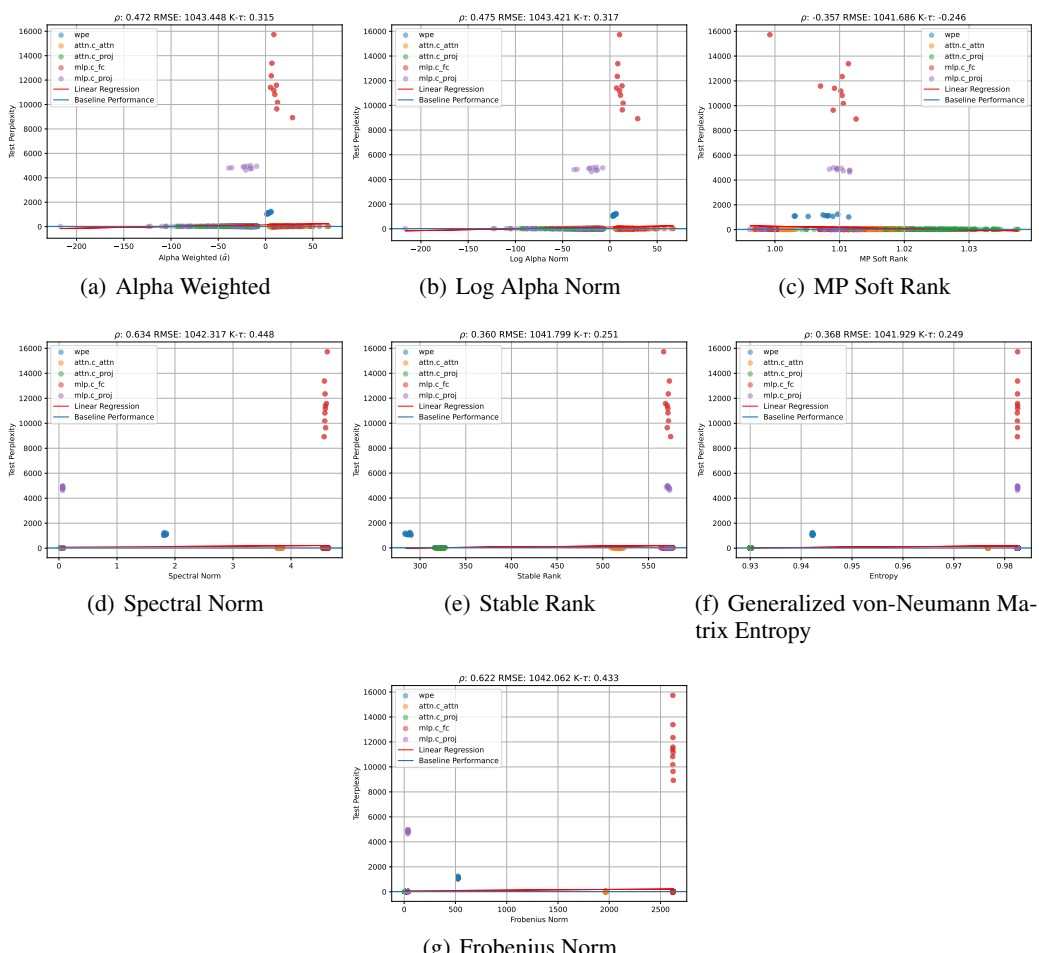

(a) Alpha Weighted     (b) Log Alpha Norm     (c) MP Soft Rank

(d) Spectral Norm     (e) Stable Rank     (f) Generalized von-Neumann Matrix Entropy

(g) Frobenius Norm

Figure 11: GPT2-Large: Layer re-randomisation test accuracy vs the respective metric. $\rho$ is the Spearman correlation coefficient, $RSME$ is the root mean square error of the linear regression (red line), and K-$\tau$ is the Kendall's tau measure, all with respect to the relationship between test accuracy and the metric.

In Figure 11, we observe the same trend that there is little to no correlation between the metric and the test accuracy of the model as found with $\alpha$ in the main body of the paper, across all metrics explored.

## B CONNECTION TO THE LOTTERY TICKET HYPOTHESIS

For our MNIST experiments, both re-initialisation and re-randomisation are drawn from the Uniform distribution, $\mathcal{U}(-\sqrt{k}, \sqrt{k})$ where $k$ is $\frac{1}{\text{in\_features}}$. However, a layer can reduce to random accuracy under re-randomisation but not under re-initialisation, which suggests that an important factor for performance retention under re-initialisation is the signs of the weights, as this represents the only difference between re-randomisation and re-initialisation.

The signs of the weights, especially within networks that use the ReLU activation, essentially represent the pathway, or subnetworks within the model, which are directly related to the Lottery Ticket Hypothesis (LTH) (Frankle & Carbin, 2018). The LTH suggests that at random initialisation, there exist subnetworks that, when trained, reaches the test accuracy of the original model that is trained with all the parameters and same compute budget (Frankle & Carbin, 2018).

To explore the role of the signs, we take the re-randomised values for a layer and apply the signs of the re-initialised layer to the re-randomised weights. We explore this using the data-based metrics that can disambiguate between re-initialised and re-randomised layers, see Figure 12. We find in Figure 12 that when applying the signs of the re-initialisation to the magnitudes of the re-randomisation layer (**green** in Figure), this layer is closer to the original layer as observed with a reduction in difference, and that there is an increase in performance when compared to the re-randomised layer (**orange** in Figure).

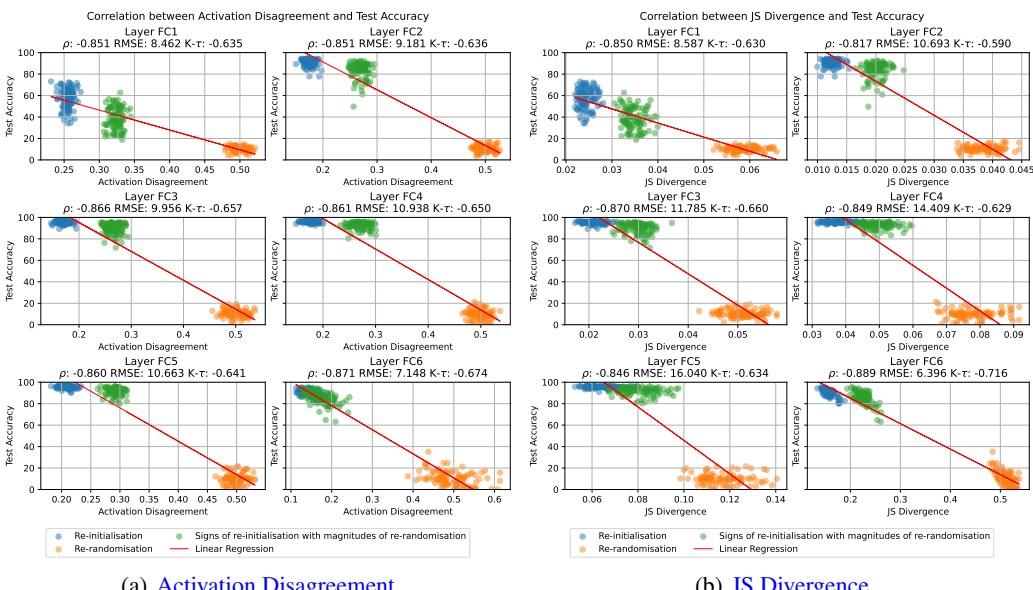

(a) Activation Disagreement        (b) JS Divergence

Figure 12: Layer re-initialisation (**blue**), re-randomisation (**orange**) and signs of re-initialisation with magnitudes of re-randomisation (**green**) test accuracy vs the respective metric. $\rho$ is the Spearman correlation coefficient, $RSME$ is the root mean square error of the linear regression (red line), and K-$\tau$ is the Kendall's tau measure, all with respect to the relationship between test accuracy and the metric.

The results strongly suggest that for this network, the subnetworks defined at the start of training through initialisation do not dramatically change through training, providing credence to the LTH. Furthermore, this demonstrates the efficacy of data-based metrics to disambiguate between the subnetwork importance of layers in neural networks, which is not captured in current data-free metrics we explore in this paper due to their inability to disambiguate robust and critical layers.

## C    FURTHER ANALYSIS OF DATA BASED METRICS

In this section, we additionally explore the Linear CKA metric's (Kornblith et al., 2019) power to distinguish between the output of a re-initialised layer from that of a re-randomised layer. We find that Linear CKA can disambiguate between a re-initialised and re-randomised layer. Although CKA has less predictive power than the Activation Disagreement and JS Divergence metrics explored in the body of the paper, as demonstrated with lower correlation values and higher RMSE, see Figure 13. These results show that not all data-based metrics can be considered equal and that there still remains a challenge in utilising the correct data-based metrics that effectively consider subnetwork relationships and disambiguate between robust and critical layers.

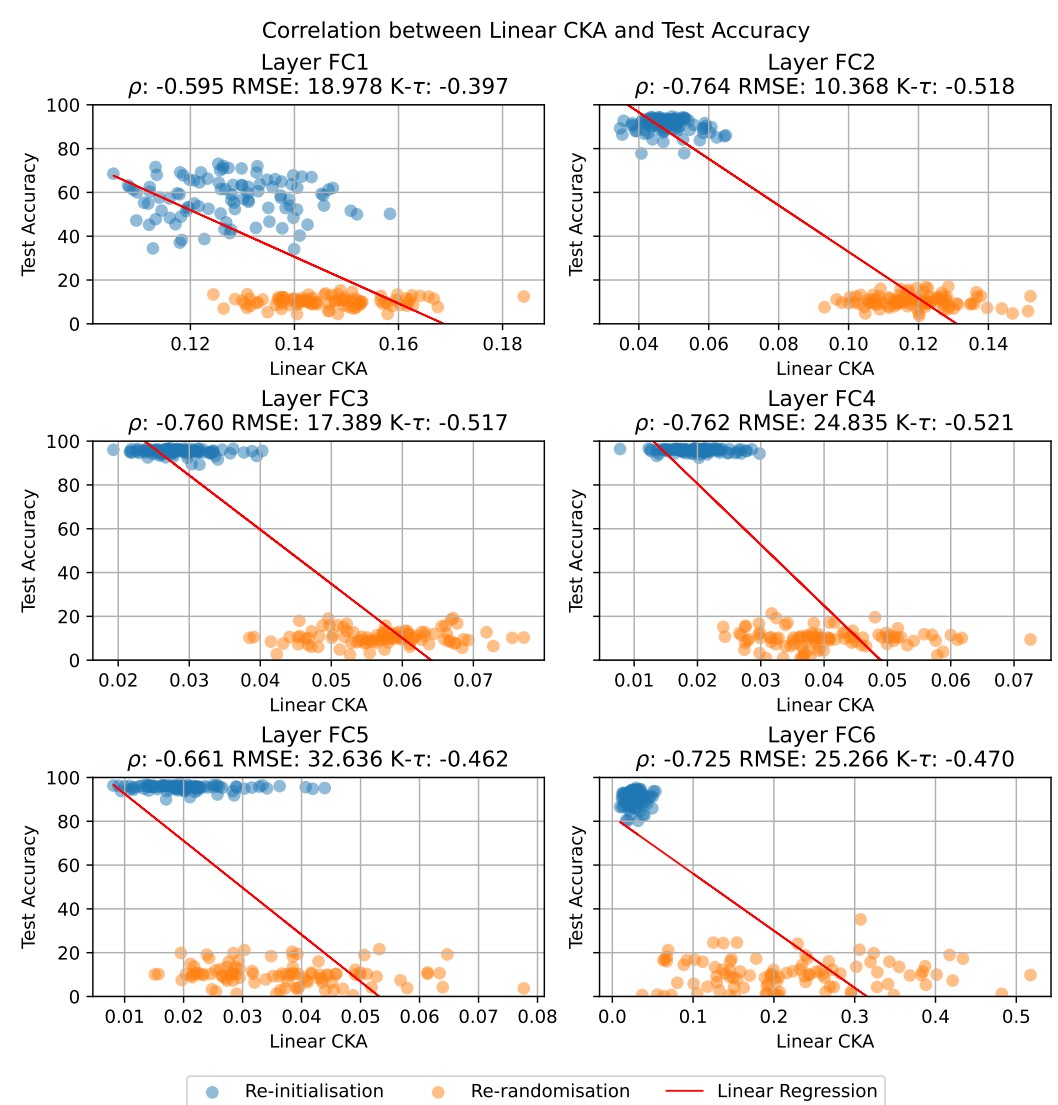

Figure 13: Layer re-initialisation (**blue**) and re-randomisation (**orange**) test accuracy vs the respective metric. $\rho$ is the Spearman correlation coefficient, $RSME$ is the root mean square error of the linear regression (red line), and K-$\tau$ is the Kendall's tau measure, all with respect to the relationship between test accuracy and the metric.

# D   LACK CAUSALITY OF ALPHA FOR GENERALISATION

We conduct the following experiments to highlight that the alpha ($\alpha$) metric is not causally related to generalisation.

Firstly, we show that a model can be initialisatised with all layers within the so-called good range ($2 \le \alpha \le 6$) and achieve random performance. Also we show that having all layers initialisatised outside of the the so-called good range ($\alpha > 6$) achieves random performance, as presented in Section D.1. Second, we show that a model can be trained with a fixed $\alpha$ values within the good range ($2 \le \alpha \le 6$), and outside of this range ($\alpha > 6$) taht achieve comparable performance. This insight which demonstrates that the $\alpha$ metric is not causally related to generalisation but is a quirk of current optimisation processes, see Section D.2.

## D.1 Alpha at initialisation

We explore the role of alpha at initialisation. Figure 2 shows that a layer can be randomly initialised within the so-called good range of $\alpha$ ($2 \leq \alpha \leq 6$). However, an $\alpha$ within this range is meant to indicate a well-trained layer.

Therefore, we explore the effect of initialising all layers with $\alpha$ within the good range ($2 \leq \alpha \leq 6$). If $\alpha$ is indicative of generalisation and a well-trained layer, then it can be expected that a model with all layers within the so-called good range should only achieve good generalisation and high test accuracy. To explore this question, we randomly sample initialisations such that we initialise a network where all the layers $\alpha$ values are within the so-called good range ($2 \leq \alpha \leq 6$) and record the test performance, we also do this for when all the layers are outside the so-called good range $\alpha > 6$ for $10 \leq \alpha \leq 12$ and $18 \leq \alpha \leq 20$ and record the test performance for a model with and without biases, see Table 4 and 5 respectively.

Table 4: MLP with biases test accuracy on MNIST with layers initialised at specific $\alpha$ values. Mean and standard error from the mean (Belia et al., 2005) derived from 100 models.

| Fix Alpha Range | FC1 | FC2 | FC3 | FC4 | FC5 | FC6 | Test Accuracy |
|---|---|---|---|---|---|---|---|
| $2 \leq \alpha \leq 6$ | $5.2452 \pm 0.0544$ | $5.2002 \pm 0.0555$ | $5.0457 \pm 0.0636$ | $5.1697 \pm 0.0542$ | $5.114 \pm 0.0668$ | $5.5775 \pm 0.0318$ | $9.9374 \pm 0.041$ |
| $10 \leq \alpha \leq 12$ | $10.8982 \pm 0.0544$ | $10.936 \pm 0.0527$ | $10.9273 \pm 0.0522$ | $10.8935 \pm 0.0535$ | $10.9935 \pm 0.0574$ | $10.9172 \pm 0.0571$ | $9.9474 \pm 0.0857$ |
| $18 \leq \alpha \leq 20$ | $19.0811 \pm 0.0519$ | $18.8586 \pm 0.0556$ | $18.9057 \pm 0.0551$ | $19.0434 \pm 0.0542$ | $18.9315 \pm 0.0552$ | $18.9897 \pm 0.0549$ | $9.9997 \pm 0.0615$ |

Table 5: MLP without biases test accuracy on MNIST with layers initialised at specific $\alpha$ values. Mean and standard error from the mean (Belia et al., 2005) derived from 100 models.

| Fix Alpha Range | FC1 | FC2 | FC3 | FC4 | FC5 | FC6 | Test Accuracy |
|---|---|---|---|---|---|---|---|
| $2 \leq \alpha \leq 6$ | $5.2452 \pm 0.0544$ | $5.2002 \pm 0.0555$ | $5.0457 \pm 0.0636$ | $5.1697 \pm 0.0542$ | $5.114 \pm 0.0668$ | $5.5775 \pm 0.0318$ | $10.0062 \pm 0.165$ |
| $10 \leq \alpha \leq 12$ | $10.8982 \pm 0.0544$ | $10.936 \pm 0.0527$ | $10.9273 \pm 0.0522$ | $10.8935 \pm 0.0535$ | $10.9935 \pm 0.0574$ | $10.9172 \pm 0.0571$ | $10.2795 \pm 0.2019$ |
| $18 \leq \alpha \leq 20$ | $19.0811 \pm 0.0519$ | $18.8586 \pm 0.0556$ | $18.9057 \pm 0.0551$ | $19.0434 \pm 0.0542$ | $18.9315 \pm 0.0552$ | $18.9897 \pm 0.0549$ | $9.7291 \pm 0.1655$ |

Table 4 and 5 show that a model can be initialised with all layers within or outside of the so-called good alpha range achieve similar accuracy (random accuracy) regardless of $\alpha$. Thereby showing $\alpha$ has no predictive power of a model's performance and is not casually related to generalisation.

## D.2 Training with any Alpha

Within WeightWatcher[6], for linear layers, the $\alpha$ metric is calculated using the squared singular values, $\Sigma^2$, of the weight matrix. Given that random initialisations can achieve a range of $\alpha$ values, including those within the so-called good range ($2 \leq \alpha \leq 6$), we sample from the random initialisation for each layer to obtain singular values that correspond to a desired $\alpha$ value, $\Sigma^\alpha$.

We then use an initialised network and decompose the layers using singular value decomposition, to obtain matrices $U\Sigma V^T$, we replace $\Sigma$ with the singular values of the desired alpha, $\Sigma^\alpha$. To obtain a new matrix, $U\Sigma^\alpha V^T$, which has the desired metric, we then keep $\Sigma^\alpha$ fixed during training, aka frozen. We then solely train $U$ and $V^T$ while maintaining the unitary properties of $U$ and $V^T$. This ensures that the model can learn, but the alpha values minimally change through training.

We explore maintaining alpha values between $2 \leq \alpha \leq 6$, $10 \leq \alpha \leq 12$, and $18 \leq \alpha \leq 20$ for all layers using the same original initialisation and only replace the singular values and show the results across 10 runs on MNIST, in Table 6 and Figure 14.

Table 6: Resulting test performance of models trained with fixed Alpha ($\alpha$) values. Mean and standard error of the mean (Belia et al., 2005) derived from 10 trained models trained on MNIST.

| Fix Alpha Range | FC1 | FC2 | FC3 | FC4 | FC5 | FC6 | Test Accuracy |
|---|---|---|---|---|---|---|---|
| $2 \leq \alpha \leq 6$ | $5.4799 \pm 0.1312$ | $5.0627 \pm 0.1209$ | $5.1361 \pm 0.1836$ | $4.9623 \pm 0.1957$ | $5.209 \pm 0.194$ | $5.4213 \pm 0.1224$ | $90.178 \pm 0.0214$ |
| $10 \leq \alpha \leq 12$ | $10.7025 \pm 0.1442$ | $11.0621 \pm 0.2103$ | $10.8441 \pm 0.1443$ | $10.8218 \pm 0.1284$ | $11.0165 \pm 0.138$ | $11.2001 \pm 0.2024$ | $90.223 \pm 0.0237$ |
| $18 \leq \alpha \leq 20$ | $19.0752 \pm 0.1244$ | $17.4638 \pm 1.2448$ | $18.8759 \pm 0.1885$ | $18.7325 \pm 0.1573$ | $18.6775 \pm 0.2023$ | $18.8356 \pm 0.1289$ | $90.219 \pm 0.0253$ |

---

[6]https://github.com/CalculatedContent/WeightWatcher

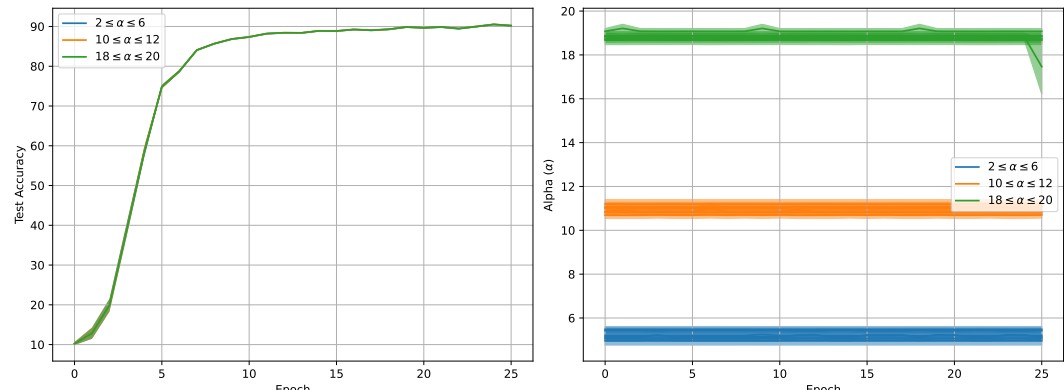

Figure 14: Test Accuracy through training with fixed Alpha ($\alpha$) on MNIST (Left). $\alpha$ value of layers thought training (Right). Mean and standard error of the mean (Belia et al., 2005) (hue) derived from 10 trained models trained on MNIST.

These results in Table 6 and Figure 14 show that the model can learn and achieve good performance regardless of the fixed $\alpha$.

All models, regardless of fixed $\alpha$, achieve similar performance. Therefore, $\alpha$ between 2 and 6 is not causally related or required for good performance. It should be noted that the model does not achieve the same performance as the main body of the paper; however, this can be attributed to fixed $\Sigma^\alpha$, which has caused a strong regularisation effect.

Furthermore, this shows that just because models generally fall within good alpha values during training, this is not a prerequisite of generalisation and more of a quirk of the current optimisation process afforded by SGD and similar variants.

