# OpenReview forum: "Efficacy of Data-Free Metrics: Robust and Critical Evidence From Robust and Critical Layers"
_ICLR.cc/2026/Conference — Submitted to ICLR 2026_

### Official Review · Reviewer_KcoL · 2025-10-31

**Soundness:** 2
**Presentation:** 3
**Contribution:** 3
**Rating:** 4
**Confidence:** 3

**Summary:**

In this paper, the authors empirically investigate the link between data-free metrics from Martin & Mahoney [1, 2] and the notion of “robust” and “critical” layers in deep neural networks. In particular, they investigate how such data-free metrics relate to reinitialisation or rerandomisation of a neural network layer’s weights. Indeed, depending on whether such a layer is “robust” or “critical”, such alterations of the parameters will respectively have little or large impact on model performance.

The authors start with a simple experimental setup using a 6-layers MLP trained on MNIST. They demonstrate that the correlation between Martin & Mahoney’s \$\alpha\$ and model performance under both reinitialisation and rerandomisation is inconsistent. Further, the range of the \$\alpha\$ does not allow distinguishing robust from critical layers. The author then shows that such findings generalise to other data-free metrics, as well as to larger scale models.

## References

[1] Charles H. Martin and Michael W. Mahoney. Implicit self-regularization in deep neural networks: Evidence from random matrix theory and implications for learning. Journal of Machine Learning Research, 22(165):1–73, 2021. URL http://jmlr.org/papers/v22/20-410.html.

[2] Charles H Martin, Tongsu Peng, and Michael W Mahoney. Predicting trends in the quality of state-of-the-art neural networks without access to training or testing data. Nature Communications, 12(1): 4122, 2021. URL https://www.nature.com/articles/s41467-021-24025-8.

**Strengths:**

I find this paper explores a very relevant topic, challenging limitations of well-established data-free metrics in determining whether the layer of a neural network is well-trained or not. Indeed, knowing the limitation of such metrics is crucial to not use them where they do not apply and draw erroneous conclusions from them.

S1: The extensiveness of the experiments provided by the authors is good. I appreciate that the paper starts with the simple example of \$\alpha\$ with a simple MLP, and then builds towards the other metrics as well as larger, common architectures such as ResNet, ViT and GPT. The additional results provided in the Appendix make the evaluation very complete.

S2: For a lot of the cases studied, the authors conclusively show that the value of the data-free metrics fails to indicate whether a layer is well- or poorly-trained after reinitialisation/rerandomisation.

S3: In addition to showing that data-free methods fail to identify the impact of reinitialisation and rerandomisation on the different layers of their models, the authors adequately demonstrate that data-driven methods can capture such changes in performance accurately. This provides a good  alternative to data-free metrics.

S4: The paper is clear, detailed and well-written. Experiments are well contextualised and related work appears to be relatively extensive.

**Weaknesses:**

Although I am convinced that this paper explores an important topic, there are some weaknesses in the experiment design which may compromise the results from the paper.

W1: I do not believe that using correlation or linear regression metrics is the right way to measure whether there is a link between \$\alpha\$ (or other metrics) and test accuracy. Referring to my question Q1, I understand from the introduction of the \$\alpha\$ metric that what matters is whether such \$\alpha\$ lies within the \$ \[ 2, 6 \] \$ range. Following-up on that, I believe the authors should evaluate the proportion of cases where \$ \alpha \in \[ 2, 6 \] \$ for layers before and after reinitialisation/rerandomisation. For example, to conclusively assess whether \$\alpha\$ has predictive power over which layer is robust or not to reinitialisation/rerandomisation, I think the authors should compare the proportion of alphas that fall within the \$ \[ 2, 6 \] \$ range before and after the operation, and also compare between reinitialisation and rerandomisation.

W2: Linked with my question Q2, I am unsure whether comparing *layer-wise* weight statistics (e.g. \$\alpha\$) to *model-wise* performance metrics (test accuracy) is a fair comparison. I believe it would be possible for a single layer to be poorly trained, but its representation to still be sufficiently aligned to conserve performance. This would allow for a bad value of \$\alpha\$ despite good overall performance.

W3: In Section 4.3, the authors use Activation Disagreement to measure differences in activations between original and modified layers. I suggest the Centred Kernel Alignment (CKA) be used instead. This metric is the state-of-the-art in measuring representational similarity between neural networks, and has the main advantage of being invariant to invertible linear transformations and as such is more robust to different initialisations.

W4: In Section 4.1, the authors mention that “an initialised, untrained layer of this network,
can fall, with a small but non-negligeable probability, within the optimal \$\alpha\$ value range of 2 and 6”. They do not study, however, if such layers lead to higher test accuracy without training. An interesting follow-up would be to assess whether such layers are more resistant to reinitialisation than layers with higher \$\alpha\$, since as per that metric, their initialisation already behaves well.

**Questions:**

Q1: In Section 2, line 91, the authors state that “a value of \$\alpha\$ between 2 and 6 as a property of a good, well-trained layer, whereas \$\alpha > 6\$ indicates that a layer is underfitted and \$\alpha < 2\$ indicates that it is overfitted”. Could the authors please clarify whether values of \$\alpha\$ matter if they fall outside the \$ \[ 2, 6 \] \$ range? For example, both a value of \$\alpha = 10\$ or \$\alpha = 1000\$ would indicate underfitting, but would that translate into strong differences in model performance? For example, for a layer which is randomly initialised, Figure 2a shows a power-law distribution of the values of \$\alpha\$ for layers whose performance should be similarly close to random guessing.

Q2: As a follow-up to the previous question, the same definition states that the value of \$\alpha\$ describes whether some specific layer is well-trained or not. On the other hand, the performance metrics used by the authors are computed at the model-level, where other layers would still have good values of \$\alpha\$ after one has been reinitialised/rerandomised. As such, it could be argued that although one specific layer is poorly trained, the neural network itself still is well trained. A poor value of \$\alpha\$ for a specific layer would thus not be incompatible with the neural network on a whole performing well. Do the authors have any elements that could contradict this claim? What does the existing literature state about this?

---

> ### Author Response · Authors · 2025-11-20
> **Response 1**
>
> Thank you for reviewing our work and identifying the core strengths. The questions raised and corresponding answers have greatly improved the paper by providing further evidence that explicitly shows that the alpha metric in question is not casually related to generalisation. In particular, highlighting that a model with all layers within the optimal alpha range can have random accuracy was especially compelling.
>
> **W1. I do not believe that using correlation or linear regression metrics is the right way to measure whether there is a link between (or other metrics) and test accuracy. Referring to my question Q1, I understand from the introduction of the metric that what matters is whether such lies within the range. Following-up on that, I believe the authors should evaluate the proportion of cases where for layers before and after reinitialisation/rerandomisation. For example, to conclusively assess whether has predictive power over which layer is robust or not to reinitialisation/rerandomisation, I think the authors should compare the proportion of alphas that fall within the range before and after the operation, and also compare between reinitialisation and rerandomisation.**
>
> Concerning the use of regression lines, we use this analysis method as it is analogous to the analysis provided to justify the use of alpha in [1]. In [1], this is shown in Figures 2 and 3 and Table 1. As a result, we show that across hundreds of models in our MLP setup, that the phenomenon of re-randomisation over re-initialisation breaks down any correlation power originally observed by the authors in [1].
>
> In Figure 2 b, it can be clearly observed that virtually all of the model layers are outside of the good alpha range of 2-6, regardless of re-randomisation and re-initialisation. This is particularly surprising as re-initialisation can preserve model performance while re-randomisation cannot. We observe no statistically significant difference for the alpha ranges for these conditions, which we would expect to see if data-free metrics were able to predict generalisation.
>
> To make this easier to assess, below we have created a table to show the proportion of models that have layers outside of the good alpha range that preserve accuracy under re-initialisation and that lose accuracy under re-randomisation such that this can be more easily understood and assessed. It is important to note that all layers fall within the good alpha value range prior to our re-intailisation and re-randomisation variants, as shown in Table 1 in the paper.
>
> Re-initialisation alpha proportions.
>
> | Alpha Value     |   FC0 |   FC1 |   FC2 |   FC3 |   FC4 |   FC5 |
> |:----------------|------:|------:|------:|------:|------:|------:|
> | below 2         |     0 |     0 |     0 |     0 |     0 |     0 |
> | between 2 and 6 |    10 |    14 |    18 |    18 |    19 |    11 |
> | above 6         |    90 |    86 |    82 |    82 |    81 |    89 |
>
> Re-randomisation alpha proportions:
>
> | Alpha Value     |   FC0 |   FC1 |   FC2 |   FC3 |   FC4 |   FC5 |
> |:----------------|------:|------:|------:|------:|------:|------:|
> | below 2         |     0 |     0 |     0 |     0 |     0 |     0 |
> | between 2 and 6 |    15 |     9 |    21 |    12 |    12 |    21 |
> | above 6         |    85 |    91 |    79 |    88 |    88 |    79 |
>
> We thank you for prompting this table. Although this can be observed from the figure, it makes the overlap between alpha values for re-initialisation and re-randomisation easier to evaluate and helps to convey our core message of the spurious correlation provided by alpha values.

---

> ### Author Response · Authors · 2025-11-20
> **Response 2**
>
> **W2. Linked with my question Q2, I am unsure whether comparing layer-wise weight statistics (e.g. ) to model-wise performance metrics (test accuracy) is a fair comparison. I believe it would be possible for a single layer to be poorly trained, but its representation to still be sufficiently aligned to conserve performance. This would allow for a bad value of despite good overall performance.**
>
> Work done by [1,2,3,4,5,6] posits that though the metrics provided via random matrix theory can infer whether a particular layer in a model is well trained, this has been used to motivate data-free: layer-wise pruning, understanding generalisation and explanations of grokking, to mention a few.
>
> As a result, given the literature's conviction in the ability of layer-wise weight based analysis to provide insights into model generalisation we argue that this is the perfect test bed to see if the predictive capacity of metrics offered by random matrix theory hold, or if they instead capture spurious correlations with generalisation.
>
> However, we show that randomly initialised untrained network layers can fall within the good alpha range of 2-6 in Figure 2a with a non-negligible probability; therefore, there naturally exists a network with good alpha values across all layers that gets random accuracy.
>
> In the table below, we show a case where a neural network has alpha values with the optimal range but still has random performance. Reported the mean and the standard error of the mean.
>
> | Fix Alpha Range | FC1     | FC2     | FC3     | FC4     | FC5     | FC6     | Test Accuracy |
> |-----------------|---------|---------|---------|---------|---------|---------|--------------------|
> | 2 and 6         | 5.2452 +- 0.0544  | 5.2002 +- 0.0555  | 5.0457 +- 0.0636  | 5.1697 +- 0.0542  | 5.114 +- 0.0668   | 5.5775 +- 0.0318 | 10.0062 +- 0.1650  |
> | 10 and 12       | 10.8982 +- 0.0544 | 10.936 +- 0.0527  | 10.9273 +- 0.0522 | 10.8935 +- 0.0535 | 10.9935 +- 0.0574 | 10.9172 +- 0.0571 | 10.2795 +- 0.2019  |
> | 18 and 20       | 19.0811 +- 0.0519 | 18.8586 +- 0.0556 | 18.9057 +- 0.0551 | 19.0434 +- 0.0542 | 18.9315 +- 0.0552 | 18.9897 +- 0.0549 | 9.7291 +- 0.1655   |
>
>
> Here, it can be observed that all mean performances, regardless of alpha values, are essentially equal due to overlapping Standard Error of the Means. These results further argue that layers with alpha values within the optimal range are not a causal factor for generalisation and are merely spuriously correlated with optimisation processes.
>
> Thank you very much for prompting us to bring this finding forward and make it more explicit, as it concretely illustrates how having a network with all layers within the good alpha range can still have random performance, and thus is not a good metric.
>
> **W3: In Section 4.3, the authors use Activation Disagreement to measure differences in activations between original and modified layers. I suggest the Centred Kernel Alignment (CKA) be used instead. This metric is the state-of-the-art in measuring representational similarity between neural networks, and has the main advantage of being invariant to invertible linear transformations and as such is more robust to different initialisations**
>
> Section 4.3 shows that there exist metrics outside of the data-free paradigm that can disambiguate between the effects of re-initialisation and re-randomisation. This shows the viability of considering the crucial interplay between weights and data, and not solely weight distributions. Given that Section 4.3 is only meant to show that this is possible, we were not exhaustive with respect to data-based metrics; however, given that the main aim of the paper is to provide evidence that refutes data-free metrics, we were exhaustive in this regard.
> We have included the results of using the Linear CKA metric and find that they largely agree with the metrics of activation disagreement and JS divergence. However, find that the Linear CKA metric has less power at distinguishing between re-initialisation and re-randomisation with lower correlation values, and higher RMSE.
>
> | Layer | Re-initlisation Accuracy | Re-initilisation CKA | Re-randomisation Accuracy | Re-randomisation CKA | RMSE | $\rho$ | $K-\tau$ |
> |---|---|---|---|---|---|---|---|
> | FC1 | 57.2214 +- 0.9356 | 0.8724+-0.0011 | 10.1519 +- 0.2175 | 0.8534 +- 0.0011 | 18.978 | 0.3997 | 0.3969 |
> | FC2 | 90.7503 +- 0.3277 | 0.9521 +- 0.0006 | 10.3883 +- 0.2726 | 0.8812 +- 0.0012 | 10.3679 | 0.9338 | 0.5177 |
> | FC3 | 95.1805 +- 0.1306 | 0.9716 +- 0.0004 | 10.5175 +- 0.3320 | 0.9430 +- 0.0008 | 17.3893 | 0.8319 | 0.5166 |
> | FC4 | 95.6963 +- 0.0804 | 0.9803 +- 0.0004 | 9.7300 +- 0.4149 | 0.9597 +- 0.0010 | 24.8352 | 0.6678 | 0.5210 |
> | FC5 | 95.3636 +- 0.1138 | 0.9797 +- 0.0007 | 10.0727 +- 0.4766 | 0.9632+-0.0012 | 32.6358 | 0.4182 | 0.4621 |
> | FC6 | 90.1141 +- 0.3321 | 0.9722 +- 0.0009 | 10.2321 +- 0.6798 | 0.7823 +- 0.0104 | 25.2658 | 0.6069 | 0.4699 |

---

> ### Author Response · Authors · 2025-11-20
> **Response 3**
>
> **W4: In Section 4.1, the authors mention that “an initialised, untrained layer of this network, can fall, with a small but non-negligeable probability, within the optimal value range of 2 and 6”. They do not study, however, if such layers lead to higher test accuracy without training. An interesting follow-up would be to assess whether such layers are more resistant to reinitialisation than layers with higher , since as per that metric, their initialisation already behaves well..**
>
> Thank you for providing the opportunity to make this point explicitly. Given that all layers are sampled from them Uniform distribution ($\mathcal{U}(−\sqrt{k}, \sqrt{k})$ where $k$ is $\frac{1}{\text{in features}}$)  randomly, the fact that these are random, untrained layers will correspond to random performance.
>
> Please see the result table provided in our response to W2. Here we compare a neural network that is initialised within the good alpha range and two neural networks that are initialised outside of the alpha range. It can be observed that there is no difference between these two conditions, showing that the good alpha range merely correlates with, rather than being a causal factor or for generalisation.
>
>
> **Q1: In Section 2, line 91, the authors state that “a value of between 2 and 6 as a property of a good, well-trained layer, whereas indicates that a layer is underfitted and indicates that it is overfitted”. Could the authors please clarify whether values of matter if they fall outside the range? For example, both a value of or would indicate underfitting, but would that translate into strong differences in model performance? For example, for a layer which is randomly initialised, Figure 2a shows a power-law distribution of the values of for layers whose performance should be similarly close to random guessing.**
>
> The statement “a value of between 2 and 6 as a property of a good, well-trained layer, whereas indicates that a layer is underfitted and indicates that it is overfitted” is based on the literature by Martin and Maloney [1,2]. They state that layers that achieve alpha between 2 and 6 are good models and that this can be used to predict overall performance [1].
>
> Our main argument is that it is important to show the correlation across the entire range because a layer's alpha value can be arbitrary, yet the model can still retain its performance.
>
> We explore how the re-initialisation and re-randomisation affect the performance of the model and record the alpha values of these layers. We find that the alpha values can fall within and out of the good and bad range; however, this has no impact on the performance. What is more important is whether the layer is a re-randomisation or re-initialisation, not the corresponding alpha value. This is why we highlight all the alpha values recorded to exemplify this point. In response to reviewer FJTL, regarding the Lottery Ticket Hypothesis, we have highlighted that a reason that re-initialization is able to maintain performance is that the model's signs are largely preserved and thus the sub-networks are maintained. This further highlights the ineffectiveness of the alpha metric or any data-free metric surveyed in the paper.
>
> The result table in our response to W2 compares a neural network initialized within the good alpha range to two neural networks initialized outside of that range. The results show no difference in performance between these conditions, indicating that being within the good alpha range only correlates with, but does not cause, generalization.

---

> ### Author Response · Authors · 2025-11-20
> **Response 4**
>
> **Q2: As a follow-up to the previous question, the same definition states that the value of describes whether some specific layer is well-trained or not. On the other hand, the performance metrics used by the authors are computed at the model-level, where other layers would still have good values of after one has been reinitialised/rerandomised. As such, it could be argued that although one specific layer is poorly trained, the neural network itself still is well trained. A poor value of for a specific layer would thus not be incompatible with the neural network on a whole performing well. Do the authors have any elements that could contradict this claim? What does the existing literature state about this?**
>
> Thank you for this question. To show that Alpha is not causally related to good performance, but is a quirk of current optimisation processes we conduct the following experiment.
>
> Within Weight Watcher (https://github.com/CalculatedContent/WeightWatcher), for linear layers, the $\alpha$ metric is calculated using the squared singular values, $\Sigma^2$, of the weight matrix. Given that random initialisations can achieve a range of $\alpha$ values, including those within the so-called good range $2-6$. We sample from the random initialisation for each layer to obtain singular values that correspond to a desired $\alpha$ value, $\Sigma^{\alpha}$.
>
> We then use an initialised network and decompose the layers using singular value decomposition, to obtain matrices  $U\Sigma V^T$, we then replace $\Sigma$ with the singular values with a desired alpha, $\Sigma^{\alpha}$, to obtain a new matrix, $U\Sigma^{\alpha} V^T$, which has the desired metric, we then keep $\Sigma^{\alpha}$ fixed during training, aka frozen. We then solely train $U$ and $V^T$ while maintaining the unitary properties of $U$ and $V^T$. This ensures that the model can learn, but the alpha values do not change through training.
>
> We explore maintaining alpha values between $2-6$, $10-12$ and $18-20$ for all layers using the same original initialisation and only replace the singular values and show the results across 10 runs on MNIST.
>
> | Fix Alpha Range | FC1 | FC2 | FC3 | FC4 | FC5 | FC6 | Mean Test Accuracy |
> |---|---|---|---|---|---|---|---|
> | 2 and 6 | 5.4799 +- 0.1312 | 5.0627 +- 0.1209 | 5.1361 +- 0.1836 | 4.9623 +- 0.1957 | 5.2090 +- 0.1940 | 5.4213 +- 0.1224 | 90.178 +- 0.0213916 |
> | 10 and 12 | 10.7025 +- 0.1442 | 11.0621 +- 0.2103 | 10.8441 +- 0.1443 | 10.8218 +- 0.1284 | 11.0165 +- 0.138 | 11.2001 +- 0.2024 | 90.223 + -0.0236664 |
> | 18 and 20 | 19.0752 +- 0.1244 | 17.4638 +- 1.2448 | 18.8759 +- 0.1885 | 18.7325 +- 0.1573 | 18.6775 +- 0.2023 | 18.8356 +- 0.1289 | 90.219 +- 0.0252765 |
>
>
> These results show that the model can learn and achieve good performance regardless of the fixed $\alpha$. All models, regardless of fixed $\alpha$, achieve similar performance. Therefore, $\alpha$ between 2 and 6 is not causally related or required for good performance.  It should be noted that the model does not achieve the same performance as the main body of the paper; however, this can be attributed to fixed $\Sigma^{\alpha}$, which has caused a strong regularisation effect.
>
> Furthermore, this shows that just because models generally fall within good alpha values during training, this is not a prerequisite of a well generalisation and more of a quirk of the current optimisation process afforded by SGD and similar variants.
>
> We will add these results into the main paper to further our arguments surrounding the unreliability of the alpha metric in Section 4.1. Thank you again for prompting this experiment which concretely demonstrates our findings on the unreliability of data-free metrics.
>
> ## References:
>
> [1] Martin, C.H., Peng, T. & Mahoney, M.W. Predicting trends in the quality of state-of-the-art neural networks without access to training or testing data. Nat Commun 12, 4122 (2021)
>
> [2] Martin, C.H. and Mahoney, M.W., 2017. Rethinking generalization requires revisiting old ideas: statistical mechanics approaches and complex learning behavior. arXiv preprint arXiv:1710.09553.
>
> [3] Lu, H., Zhou, Y., Liu, S., Wang, Z., Mahoney, M.W. and Yang, Y., 2024. Alphapruning: Using heavy-tailed self regularization theory for improved layer-wise pruning of large language models. Advances in neural information processing systems, 37, pp.9117-915
>
> [4] Lu, H., Zhou, Y., Liu, S., Wang, Z., Mahoney, M.W. and Yang, Y., 2024. Alphapruning: Using heavy-tailed self regularization theory for improved layer-wise pruning of large language models. Advances in neural information processing systems, 37, pp.9117-915
>
> [5] Qing, P., Gao, C., Zhou, Y., Diao, X., Yang, Y. and Vosoughi, S., 2024. Alphalora: Assigning lora experts based on layer training quality. arXiv preprint arXiv:2410.10054.
>
> [6] Prakash, H.K. and Martin, C.H., 2025, June. Grokking and Generalization Collapse: Insights from HTSR theory. In High-dimensional Learning Dynamics 2025.

---

> > ### Comment · Reviewer_KcoL · 2025-11-26
> > **Rebuttal follow-up**
> >
> > I would like to thank the authors for their complete and detailed rebuttal. I have taken some time to read the Nature Communications version of Martin & Mahoney’s work [1] in more detail, please note that my mentions hereinafter refer to that version. To avoid confusion when referencing, I use the notation Fig Martin.1 to designate Figures from [1], in order to not mix them up with Figures from the authors’ submission.
> >
> > [1] Charles H Martin, Tongsu Peng, and Michael W Mahoney. Predicting trends in the quality of state-of-the-art neural networks without access to training or testing data. Nature Communications, 12(1): 4122, 2021. URL https://www.nature.com/articles/s41467-021-24025-8.
> >
> > **W1:** I now recognise that Martin & Mahoney also used least squares regression in their work to show the link between data-free metrics and test accuracy. Since the authors aim to replicate that experimental setup to robust and critical layers, my remark suggesting another way to evaluate these experiments does not make sense. I therefore consider this point addressed.
> >
> > **W2/Q2:** I am not convinced by the authors’ argumentations about using a layer-wise statistic to predict the accuracy of the entire network. From my understanding of [1], regression analyses such as the one in Table Martin.1 are computed using model-wise statistics, such as weighted alpha $\hat{\alpha}$ as defined in Equation Martin.(10) (and not the one defined in Equation Martin.(5), it is confusing to me that they use the same notation but I am pretty convinced they use the model-averaged version). My understanding of the authors’ submission is that they use $\alpha$ / other layer-wise metrics as the independent variable of their regression analysis, is that correct?
> >
> > In [1], Martin & Mahoney show several examples where individual layers can have a poor value of $\alpha$, while the model as a whole maintains high performance. For example, regarding Fig Martin.5b, Martin & Mahoney write it “*depicts two very large $\alpha \gg 6$ values for the baseline [...]. This suggests the baseline model has at least two over-parameterized/under-trained layers*”, while reporting accuracy of $91.45\%$. As such, I believe it is not true to state that Martin & Mahoney claim that bad values of $\alpha$ for a single / a few layers should necessarily be linked to a generally poor performance of the entire model. It follows that a single layer-wise $\alpha$ being high will not necessarily predict model performance. The authors show this in this submission, but from my understanding, this does not contradict Martin & Mahoney’s claims.
> >
> > Furthermore, I think the $\alpha$ values do work quite well for what they are intended for: differentiating trained vs. undertrained layers; with the caveat identified by the authors that for some randomly initialised layers, $\alpha$ still falls between 2 and 6 with small probability. As shown in Fig.4, with regard to robust vs. critical layers, what really matters is whether the representation after the re-initialised/re-randomised layer remains similar/well-aligned. While I agree with the authors that Martin & Mahoney’s metrics mostly fail at predicting this, I think this falls out of the scope of what these metrics are meant to measure.
> >
> > **W3:** Thank you for running this experiment. I find it very interesting that linear CKA is less predictive of re-initialisation / re-randomisation than the other data-based metrics studied by the authors. I would be very curious to understand why, but I think this falls outside the scope of what can reasonably be asked from the authors in the scope of the discussion period. I agree with  the authors that the other two metrics effectively demonstrate the effectiveness of data-based metrics, but I encourage them to include the CKA analysis as an Appendix. I therefore consider this point partially addressed.
> >
> > **W4:** I understand that it is possible for randomly initialised layers, with relatively small probability, to have an $\alpha \in \[2, 6\]$, without it being meaningful with regard to the test accuracy of the model they are a part of. I consider this point addressed.
> >
> > **Conclusion:** More generally speaking with regard to the link between the authors’ submission and Martin & Mahoney’s work [1], the authors do effectively demonstrate that there is no causal link between individual layer-wise values of $\alpha$ - eventually across multiple or all layers - and test accuracy. From reading [1], I do not see Martin & Mahoney making any claims of a causal link; I think they rather show strong empirical correlation. I do not think that the existence of some edge cases identified by the authors where this method does not hold invalidates Martin & Mahoney's work or their metrics, which remain valid for most practical cases.

---

> > > ### Author Response · Authors · 2025-11-27
> > > **Rebuttal follow-up author response (1/2)**
> > >
> > > **Overall:**
> > >
> > > We appreciate your commitment to the review process and taking the time to review our responses to your weaknesses and questions - we felt that your line of questioning was particularly helpful for adding improved strength to our findings and analysis.
> > >
> > > **W1:**
> > >
> > > We are pleased that we were able to provide clarity on this point and show that our use of evaluation metrics for correlations mirrored that of the original analysis of [1].
> > >
> > > **W2:**
> > >
> > > Thank you for your thorough review of [1]. Could you clarify why you believe these metrics effectively differentiate well-trained from poorly trained layers, given that the empirical evidence in our paper demonstrates they do not?
> > >
> > > Furthermore, we would like to highlight that it is not that data-free metrics “mostly fail” in distinguishing between robust and critical layers. Data-free metrics fail entirely at predicting this. As a result, it is a logical fallacy to state that such metrics are useful for identifying a well-trained layer. Following our analysis, it is more apt to conclude that they provide a spurious correlation with well-trained layers that can lead to misleading insights based on such erroneous relations.
> > > Alpha and other data-free metrics are intended to identify well-trained versus poorly trained layers. Our response to Q2 clearly demonstrates that, under identical computational budgets, networks optimised to be inside and outside the good alpha range perform equivalently. Therefore, what can their analysis tell us in traditional settings that one cannot understand from the accuracy of the model?
> > >
> > > Our results irrefutably show that these metrics fail to predict well-trained versus poorly trained layers and that they do not predict well-trained layers or networks. Can you explain why, in light of this evidence, you continue to argue that data-free metrics have the properties of metrics that can predict well-trained versus poorly trained layers and/or networks when we show repeatedly that they cannot?
> > >
> > > **W3:**
> > >
> > > We appreciate your recognition of our additional analysis considering CKA., We agree that it is interesting that it has less predictive capacity than the other data-based metrics. We would reasonably presume that, as we demonstrated in our response to reviewer FJTL Q2, CKA does not consider subnetworks of the model to the same degree as the other metrics we have explored. However, as you have stated, confirming this is out of scope.
> > > To fully address this point, we will add an appendix section containing the results from CKA. This will be in the revised paper that we will upload before the end of the rebuttal period.
> > >
> > >
> > > **W4:**
> > >
> > > We are very glad that our experiments highlighted that an entire network can have the optimal Alpha value and still achieve random accuracy - showing effectively that alpha is spuriously related with generalisation rather than being a causal factor for it.

---

> ### Author Response · Authors · 2025-11-27
> **Rebuttal follow-up author response (2/2)**
>
> **Conclusion**
>
> Our paper does not and cannot provide an exhaustive analysis of all the ways that these data-free metrics can fail, but instead provides a suitable case that shows they are not reparameterization invariant. It is highly likely, given our results, that there are additional failure cases of these methods that have yet to be identified. For example, we show this with an experiment (in response to your Q2), which shows a model can generalise even with fixed alpha during training. Therefore, it can be safely assumed that under different training paradigms or optimisation methods, alpha and other data-free metrics are likely to fail.
>
> We appreciate your acknowledgement that neither individual layer-wise nor entire network alpha values are causally linked to test accuracy (generalisation), which is a key metric offered by Random Matrix Theory.
>
> It is of great scientific importance to understand what properties are causal for neural network generalisation.
> Random Matrix Theory has been proposed as a strong candidate for explaining why generalisation occurs [2]. Therefore, it is essential for the community to determine whether the metrics it provides are truly causally important for generalisation.
>
> There is a strong precedence in literature for studying if properties that strongly correlate with generalisation such as flat minima are casual factors for generalisation - the flat minima work by Dihn et al [3] that showed flat minima were not causal despite being strongly correlated  with generalisation changed the field and has been cited over 900 times for doing so (**we kindly as that you read our responses to reviewer FJTL W3 where we elaborate on this further**).
> Our work demonstrates that, as you have acknowledged based on our empirical evidence, data-free metrics from RMT are not causal to generalisation.
>
> Furthermore, our analysis shows that practical applications of these data-free metrics can be misleading and may result in erroneous conclusions. As such, their value to the community is limited. **It important for data-free metrics to consider the range of negative cases we provide to push forward the discovery of data-free metrics that could be casual to generalisation.**
>
> Provided this is the International Conference on Learning Representations, it is essential to communicate that weight space learning representations (isolated weight space analysis) cannot (under existing data-free metrics) be analysed in isolation to causally understand generalisation. This is the insight provided by this work, beyond any assertions from [1] and the literature that has preceded it (**please see our response to reviewer VRWM W1 for a full description of work that preceded 1, which argues such data-free metrics are important for understanding generalisation**).
>
> **References:**
>
> [1] Charles H Martin, Tongsu Peng, and Michael W Mahoney. Predicting trends in the quality of state-of-the-art neural networks without access to training or testing data. Nature Communications, 12(1): 4122, 2021. URL https://www.nature.com/articles/s41467-021-24025-8.
>
> [2] Wei, A., Hu, W. and Steinhardt, J., 2022, June. More than a toy: Random matrix models predict how real-world neural representations generalize. In International conference on machine learning (pp. 23549-23588). PMLR.
>
> [3] Dinh, L., Pascanu, R., Bengio, S. and Bengio, Y., 2017, July. Sharp minima can generalize for deep nets. In International Conference on Machine Learning (pp. 1019-1028). PMLR.

---

> > ### Comment · Reviewer_KcoL · 2025-11-27
> > **Rebuttal follow-up #2**
> >
> > I thank the authors for their quick response. Please find some clarifications from my side below.
> >
> > **W2**
> >
> > > Could you clarify why you believe these metrics effectively differentiate well-trained from poorly trained layers, given that the empirical evidence in our paper demonstrates they do not?
> >
> > Figure 2.a shows the distribution of $\alpha$ values for untrained layers (initialisations), and Table 1 shows the same distribution for trained layers. It appears quite evidently that these distributions are not the same, and that trained layers show values $\alpha < 5$. As per the density plot of Fig2.a, around $5\\%$ of untrained layers have $\alpha < 5$. As such, I feel it is reasonable to state that $\alpha$ values are generally good indicators of whether a layer is trained. In their rebuttal, “Response 4”, the authors demonstrate a method to adversarially generate NNs that maintain good accuracy despite higher $\alpha$ values. While this shows the absence of a causal link between $\alpha$ and test accuracy, I think this adversarial example goes beyond the scope in which one can reasonably expect to use data-free metrics in practice.
> >
> > > Furthermore, we would like to highlight that it is not that data-free metrics “mostly fail” in distinguishing between robust and critical layers. Data-free metrics fail entirely at predicting this.
> >
> > From [1], “HT-SR Theory predicts that smaller values of $\alpha$ should correspond to models with better correlation over multiple size scales and thus to better models”. Why would the data-free metrics be capable of distinguishing between robust and critical layers under re-initialisation, since in that case weights are reset to random values? I believe this concern is shared by reviewer VRWM (see their W1), and I am not convinced by the authors’ rebuttal to that weakness.
> >
> > **Conclusion**
> >
> > > Our paper does not and cannot provide an exhaustive analysis of all the ways that these data-free metrics can fail, but instead provides a suitable case that shows they are not reparameterization invariant.
> >
> > I believe there is a confusion throughout the submission about what the data-free metrics are supposed to measure. Those metrics are supposed, per [1], to discriminate between “well-trained versus poorly trained models”. In their experimental setup, the authors use either re-initialisation or re-randomisation. Both these operations replace a layer’s weight with random initialisations. As such, they replace a *trained* layer with an *untrained* layer. Consequently, the $\alpha$ value of these layers follows the distribution in Fig2.a. Since in both re-initialisation and re-randomisation the weights are random, I do not see why one would expect their $\alpha$ to be distributed differently, and why the authors hypothesise that it should be reparametrisation invariant. What the authors then measure is whether the representation after these layers are still similar, with higher similarity yielding higher accuracy (Sec. 4.3). As per my previous remarks, a single/a few poorly trained layers within the model are not incompatible with the model as a whole performing well.
> >
> > > It important for data-free metrics to consider the range of negative cases we provide to push forward the discovery of data-free metrics that could be casual to generalisation.
> >
> > While I agree with that statement in general, I unfortunately disagree that this is the main claim of this submission or that it is achieved by its results. The contributions, as mentioned in the introduction of the submission, are to show that “data-free metrics are not invariant under reparametrisation”, that “data-free analyses have no predictive capacity to identify robust and critical layers” and that “norm-based metrics have no predictive capacity over the change in performance under re-randomisation”. As discussed, Martin & Mahoney’s data-free metrics were designed to distinguish between well-trained and poorly trained layers, not to evaluate invariance under reparameterisation or identify robust vs. critical layers. Thus, I do not see why these metrics should be expected to perform the tasks explored in this submission.
> >
> > For these reasons, I am unfortunately unwilling to recommend an “Accept”. I think that the general goal pursued by the authors, to demonstrate empirical limitations to Martin & Mahoney’s data-free metrics, remains a very interesting research direction. I do agree that understanding limitations of such metrics is important. However, I am not convinced that the particular example of analysing robust vs. critical layers is relevant to those metrics, as this use-case falls outside their intended scope.

---

> ### Author Response · Authors · 2025-11-27
> **Rebuttal follow-up #2 author response (1/2)**
>
> We thank the reviewer for their swift response and engagement in the rebuttal process, this deeply appreciated.
>
> `P1. While this shows the absence of a causal link between  and test accuracy, I think this adversarial example goes beyond the scope in which one can reasonably expect to use data-free metrics in practice.`
>
> Again, we appreciate that you acknowledge that alpha and data free metrics are not causal to generalization.
>
> We believe that the disagreement here stems from what it fundamentally means to have a well trained layer. Data-free literature would assert that it would be a layer with an alpha value between 2 and 6. Our example, while adversarial, shows that this cannot be the case.
>
> The experiment you requested to show that a network can have all layers with values outside of the alpha range that still generalise further shows that alpha is not important.
>
> Therefore using alpha in practice loses its utility as it 1) It is not causally important for generalisation. 2) The definition of a well trained layer being between the range of 2-6 does not hold, questioning the entire notion of what quantifies a well trained layer.  3) An untrained layer with non-zero probability can have an alpha value of 2-6 completely contradicting the idea, or rather definition, of a ‘well-trained’ layer (as we show in our responses).
>
> Also we would like to highlight that we explore a range of data-free metrics: Alpha, Alpha Weighted, Log Alpha Norm, MP Soft Rank, Spectral Norm, Stable Rank, Generalized Von-Neumman Matrix Entropy and Frobineus Norm. As a result, this paper is not solely focused on [1], however, we do highlight [1] and alpha purely because of its specific definitions of a well trained layer, which we show is false.
>
> `P2: As such, they replace a trained layer with an untrained layer. Consequently, the  value of these layers follows the distribution in Fig2.a. Since in both re-initialisation and re-randomisation the weights are random, I do not see why one would expect their  to be distributed differently.`
>
> You are correct that the distributions are the same, and that this is expected. However, as shown by the critical and robust layer work, the two similarity distributed representations have wildly different impacts on generalization, with re-initialisation preserving accuracy and re-randomisation destroying accuracy.
>
> We align this to the Lottery Ticket Hypothesis as prompted by reviewer FJTL which explains why this occurs due to subnetwork preservation, something which current data free metrics do not consider.
>
> Additionally, it was of interest to you to see whether the CKA metric could differentiate between the robust and critical layers. Our results showed that although CKA could differentiate, the power of this metric was weaker than the metrics we compared to the data-based metrics in the paper. Again, you were interested to understand why this was the case (although outside of the scope of this work). Following this line of thinking, we  do not see how understanding CKA’s weakness is different from understanding if the data-free metrics could disambiguate between robust and critical layers. For example, if we had identified a data-free metric that could differentiate, it would have been equally as interesting.  Moreover, with respect to understanding the distinction between metrics that can and cannot differentiate we provide an answer with the connection to the Lottery Ticket Hypothesis.
>
> To reiterate some results that we presented to you, we show that the similar distributions of re-initialisations and re-randomisation can have the same probability of having an alpha in a good range of 2-6, as a result we ask: Given alpha is” designed to distinguish between well-trained and poorly trained layers” how is this compatible with our findings that show untrained (random) layers can and do exist within the good alpha range of 2-6?

---

> ### Author Response · Authors · 2025-11-27
> **Rebuttal follow-up #2 author response (2/2)**
>
> `P4: “unwilling to recommend an “Accept””`
>
> As stated in your previous response we were able to 3 out of 4 (75%)  of your concerns with our work (with W3 being fully addressed with our new appendix section which will be uploaded with the revised version) - we do not understand how addressing these concerns (especially those requiring empirical experiments) have not improved the quality of this paper unless they were not important concerns related to assessing the quality of the paper.
>
> Could you please provide an explanation for why addressing these weaknesses has not improved the paper?
>
>
> `P3: “this use-case falls outside their (data-free metrics) intended scope.”`
>
> Politely we disagree. Without our extensive study it would not be obvious that the alpha value range of 2-6 is not causally important to generalisation given its very strong correlation. This is especially important as a growing body of work fundamentally relates RMT properties to fundamental aspects of generalisation such as understanding grokking and delayed generalisation [2].
>
> Additionally a fundamental and critical aspect of science is understanding what is merely a correlation and what is causally relevant to improving understanding. In this case we  provide robust and critical evidence to show that current data-free metrics do not currently provide any causal understanding of neural networks. Which requires the community to rethink how we attempt to analyze and understand these systems and the metrics we use to do so.
>
> References:
>
> [1] Charles H Martin, Tongsu Peng, and Michael W Mahoney. Predicting trends in the quality of state-of-the-art neural networks without access to training or testing data. Nature Communications, 12(1): 4122, 2021. URL https://www.nature.com/articles/s41467-021-24025-8.
>
> [2] Prakash, H.K. and Martin, C.H., 2025. Grokking and Generalization Collapse: Insights from HTSR theory. arXiv preprint arXiv:2506.04434.

---

> > ### Comment · Reviewer_KcoL · 2025-11-27
> > **Rebuttal follow-up #3**
> >
> > I appreciate the authors’ detailed rebuttal and the additional experiments, which have addressed several of my concerns. However, I remain unconvinced that the core issue has been resolved. The paper’s focus on robustness under re-initialisation and re-randomisation, while interesting, still, in my opinion, falls outside the intended scope of Martin & Mahoney's metrics. As this is the core research question of the paper, I maintain my original recommendation and score.

---

> > > ### Author Response · Authors · 2025-11-27
> > >
> > > Thank for you thoroughly engaging with the review process.
> > >
> > > We would like to make clear that we do think this analysis falls within the intended scope of Martin & Mahoney's metrics.
> > >
> > > To demonstrate this we highlight some claims that are on weightwatcher (the tool we used to perform the metric analysis) website (https://weightwatcher.ai/), which is owned by Calculation Consulting, which is owned by Charles H. Martin one of the predominate authors of the Heavy-Tailed Self-Regularization theory that put forwards the alpha metric.
> > >
> > > First the website introduces itself as the following `“WeightWatcher (w|w) is an open-source, diagnostic tool for analyzing Deep Neural Networks (DNN), without needing access to training or even test data. It is based on theoretical research into Why Deep Learning Works, relying on the older phenomenology theory of Heavy-Tailed Self-Regularization (HTSR), published in JMLR, Nature Communications, etc., and the more recent SemiEmpirical Theory of (Deep) Learning (SETOL), presented at NeurIPS2023.”`
> > >
> > > This suggests that the metrics (specfically alpha) that they have come up with are based on theoretical understanding as `“Why Deep Learning Works”`, we show that these metrics are not required for model generalisation and does not effectively explain `“Why Deep Learning Works”` as it is not causally related to model performance.
> > >
> > > The website goes on to state the the 5 main use cases of the metrics are as follows:
> > >
> > > 1. `"Identify poorly trained layers in your models"`
> > > 2. `"Help you select the best pretrained model"`
> > > 3. `"Predict (trends in) test accuracies -- without training or test data"`
> > > 4. `"Evaluate the information flow in different architectures"`
> > > 5. `"Find (and remove) training anomalies"`
> > >
> > > Given how the author, markets their research this suggests that it is fair to assume that there work could have potentially identified robust/critical layers within a neural network, given they state it can do 1,2,3,4 and 5. However, we provide robust and critical evidence that alpha, cannot and does not indicate a well trained layer and that and that alpha between 2 and 6  is not causally required for a model to perform well. Which goes directly against claims 1,2,3,4 and 5. Claim 4 states that the metric can be used to understand understand the information flow within a neural network, we empfasticly show this is not the case by demonstrating that layers can have arbitrary alpha’s with varying performance, therefore this metric cannot demonstrate the information flow and how well the information flows through the network.  Claim 5, suggests that the method can identify training anomalies, a layer that can be randomised without harming performance, thus could random from the beginning of training, can be considered a training anomaly as it does not require training this suggests these data-free metrics are unable to effectively do this.
> > >
> > > We would like to make clear that this work approach this investigation with the understanding that the data-free metrics offered by Martin and Maloney could potentially identify robust and critical layers. If these metrics could have identified the robust and critical layer phenomena, then this would have given additional credence to the metrics and theory provided by Martin and Mahoney. However we found that it could not, furthermore that a model can be initialised with alpha values within the good and bad range, and thanks to your questions demonstrated that a model can learn to generalise with any alpha.
> > >
> > > For the above reasons we firmly disagree that our study is outside of the scope of the metrics and the claims of the work that has extended on the original work.

---

### Official Review · Reviewer_FJTL · 2025-11-01

**Soundness:** 2
**Presentation:** 2
**Contribution:** 1
**Rating:** 4
**Confidence:** 2

**Summary:**

:
This paper studies data-free metrics, primarily those based on random matrix theory from Martin and Mahoney, 2021. The authors find that these metrics cannot reliably disambiguate between robust and critical layers which have been measured by model’s performance under re-initialization and re-randomization.  The paper questions causality though I believe that it is not the claim of the prior work.
In contrast they also look at data-based metrics (activation disagreement and Jensen-Shanon divergence of the softmax) and show that as more plausible metrics.

**Strengths:**

- The experimental framework is well-designed. Using re-initialization vs. re-randomization provides clear functional ground truth for layer importance.The paper is covering a wide range of architectures (ResNet, ViT, GPT-2).
- The paper explicitly tests claims from Martin & Mahoney (2021), making it easy to assess.
- Strong evidence where alpha (and other metrics) shows no correlation with test accuracy.

**Weaknesses:**

- The paper is purely evaluative, showing what doesn't work without offering novel alternatives beyond briefly looking at data-based metrics without further insights or extensive evaluation.
- The paper demonstrates that correlations fail but doesn't explain why.
- The authors briefly mention the flat minima literature (Dinh et al. 2017\) however the connection is not clear in the paper.

**Questions:**

- While I briefly checked Martin and Mahoney for this review, it is unclear to me if expecting alpha to stay in the same range after re-initialization or re-randomization is plausible. Given the density of Martin and Mahoney’s work the underlying assumptions and why the authors would expect alpha to be concentrated should be explained better in the paper.
- I realize they are not directly comparable, but do the authors have insights connecting their work to the Lottery Ticket Hypothesis? The re-initialization results (in data-based methods mostly) seem related to findings about network pruning and subnetworks. This could help situate the findings in the broader interpretability literature.
-  While the call to explore alternative methods is valuable, do the authors have concrete recommendations?

---

> ### Author Response · Authors · 2025-11-20
> **Response 1**
>
> Thank you for reviewing our paper, appreciating the well-designed experiments and highlighting its strengths in showing that alpha and other data-free metrics have no correlation with test accuracy, which is an important and timely finding. The questions and weaknesses identified have helped us better situate this work and improve the paper. The question regarding the lottery ticket hypothesis was particularly insightful and we look forward to adding these results to the main body of the paper.
>
>
> **W1. The paper is purely evaluative, showing what doesn't work without offering novel alternatives beyond briefly looking at data-based metrics without further insights or extensive evaluation.**
>
> Given the breadth and depth of literature focusing on the use of data-free metrics for interpretability of neural networks, we felt that it would be most compelling to provide the most robust empirical evidence possible that refutes the predictive power of data-free metrics across model complexities, datasets and data modalities. The novel offering of our work is that it provides such strong evidence against the use of data-free methods.
>
> There are many precedents in the ML literature of empirical/evaluative work that forced us to rethink commonly held beliefs. For example:
>
> 1. The seminal work on the loss landscape geometry by Dhin et al (2017) [1], which showed that hessian based sharpness measures were not reparameterisation invariant and, therefore, that flat minima were correlated with generalisation but not a causal aspect of them. [1] fundamentally changed the way people consider loss landscapes and has been cited 996 times for its presentation of negative results for flat minima. This paper also did not offer any alternative solutions or methods to handle this issue.
>
> 2. Work showing that neural networks can fit random data [2] provided fundamental new insights into the overparameterisation of neural networks. While this work offered no way to remove memorisation in neural networks, it has been highly impactful in considering generalisation.
>
> 3. The original paper on grokking [3] provides another example of evaluative work that empirically demonstrates a phenomenon (of severely delayed generalisation in neural networks for modular arithmetic task), but provided no further insight (e.g. a solution for overcoming the large memorisation phase during training on these tasks). It nevertheless opened a whole new research avenue.
>
> Our work provides a similar empirical/evaluative contribution, providing solid empirical evidence that we must rethink neural networks beyond considering the heavy-tailed distributions of weight spaces, which are prioritised by data-free metrics [4,5,6,7,8,9] and, instead, focus on interactions between data and weights as shown by our data-based metric analysis.
>
> This being said, our work goes beyond the kind of purely evaluative contribution that can be found in the seminal papers [1,2,3]. It offers an alternative to data-free metrics by showing how to predict critical and robust layers employing data-based metrics. This suggests a new avenue to explore these methods further and improve our understanding of neural networks.
>
> We hope that our work can push the field forward in two core ways.
>
> 1. Providing a strong test ground for the development of data-free metrics that are able to disambiguate between robust and critical layers. Similar to how the work of Dhin et al (2017) [1] led to the development of sharpness metrics Relative Flatness [10] and Fisher Rao norm [11] that are reparameterisation invariant and now considered state of the art sharpness metrics as a result.
>
> 2. Concrete evidence for prioritizing data-based metrics, until the appropriate data-free metrics are discovered , such as those that we examine in the paper to improve efforts in interpretability and compression.
>
> We will be sure to highlight this in the main body background section such that this is an explicit contribution of our work rather than implicit.

---

> ### Author Response · Authors · 2025-11-20
> **Response 2**
>
> **W2. The paper demonstrates that correlations fail but doesn't explain why.**
>
> Thank you for providing us with the opportunity to further explain this point.
>
> We argue that the correlations of the alpha (and other data-free) metrics fail because they do not capture the nuanced interplay between both the data passed through the model and the weights that process it.
>
> By focusing solely on the weight distribution of parameters, any reparameterisation of these layers (such as re-randomisation or re-initialisation) will fundamentally change the value of any data-free metric, even in the event that such reparameterisation does not alter the performance of the model. Moreover, these correlations fail for data-free metrics because they are fundamentally flawed and can only provide a spurious correlation of metrics of interest such as generalisation. As such, they should be abandoned for more holistic (data-based) metrics that consider neural networks beyond the single dimension of weight distributions.
>
> In addition please see our response to your question regarding the Lottery Ticket Hypothesis, where we show that the core reason why data-free metrics fail in the case of robust and critical layers is because they do not acknowledge the importance of sub networks. In turn, through this experiment you requested we were able to show why data-based metrics do work as they consider this fundamental aspect of internal computation. This cannot be accounted for by current data-free metrics which study weights in isolation [4,5,6,7,8,9].
>
> **W3. The authors briefly mention the flat minima literature (Dinh et al. 2017) however the connection is not clear in the paper.**
>
> We apologise that the connection to the seminal work provided by Dinh et al. 2017 was not clear. Please see our response to W1 where we discuss this in relation to other seminal literature that provide interesting yet negative results.
>
> We argue that our findings serve a similar purpose to other seminal literature in the loss landscape geometry by Dhin et al (2017) [1]. In this work Dhin et al provided strong evidence displaying that hessian based sharpness measures were not reparameterisation invariant. As a result of this, new insights were delivered to the field, flat minima were correlated with generalisation but not a causal aspect of them. This work also led to the development of sharpness measures which were then invariant to reparameterisation such as Relative Flatness [10] and Fisher Rao norm [11].
>
> Similarly, we empirically show through the use of re-initialisation and re-randomisation across architectures and complexity scales that data-free metrics are fundamentally flawed in predicting generalisation due to their inability to distinguish between the wildly varying performance differences offered by these two reparameterisations. As a result, we argue that if it is possible, data-free metrics should be pursued that are able to distinguish between these controls, but if not, that data-free based interpretability should not be pursued over data-based metrics.
>
> **Q1. While I briefly checked Martin and Mahoney for this review, it is unclear to me if expecting alpha to stay in the same range after re-initialization or re-randomization is plausible. Given the density of Martin and Mahoney’s work the underlying assumptions and why the authors would expect alpha to be concentrated should be explained better in the paper.**
>
> The motivation of our work is that we do not expect alpha to remain within the same range under re-intailisation and re-randomisation, as re-intialisation can preserve performance we would expect the alpha to be in the good range but that tre-randomisation would be outside the good range because it massively degrades performance.
>
> However, what we find is that there is no statically significant difference in the distribution of alpha values for these two conditions despite stark differences in generalization offered by these two conditions.
>
> Given statements surrounding the alpha, and other data-free, metrics capability to predict the performance of a model based on weight analysis alone [4,5,6,7,8,9] we would expect that it can distinguish between robust and critical layers if the alpha value is measuring something that is causally related to a model performing well.
>
> However, across datasets, architectures and modalities we find that the alpha metric and other data free metrics are incapable of this as when layers are outside of the optimal range under re-randomisation or re-initialisation performance can still be retained. Therefore, our work serves a strong negative result for the importance of any particular layer quality that is captured by the alpha metric.

---

> ### Author Response · Authors · 2025-11-20
> **Response 3**
>
> **Q2. I realize they are not directly comparable, but do the authors have insights connecting their work to the Lottery Ticket Hypothesis? The re-initialization results (in data-based methods mostly) seem related to findings about network pruning and subnetworks. This could help situate the findings in the broader interpretability literature.**
>
> Thank you for this insightful question. We do think that the insights from our work are directly related to the Lottery Ticket Hypothesis (LTH) and have provided results that concretely highlight this - we thank you for prompting this as it adds to the depth of our contribution.
>
> Given that both re-initialisation and re-randomisation are drawn from the Kaiming Uniform distribution, but that a critical layer can lose accuracy altogether under re-randomisation but not under re-initialisation it suggests that a considerable factor for performance retention under re-initialisation is the signs of the weights.
>
> The signs of the weights, especially within networks that use the ReLU activation, essentially represent the pathway, or subnetworks within the model, which are directly related to the LTH.
>
> To further elucidate this point, we have provided a further experiment that takes re-randomised values for a layer and applies re-initialised signs of the re-initalised layer to the re-randomised weights.
>
> When we analyse this with the data-based metrics we explore in the paper, we find that the distance between re-initialisation and re-randomisation is massively reduced and that the performance is largely recovered.
>
> For ease of evaluation we provide a Table with the activation disagreement distances under the three conditions: Re-initialisation, Re-randamisation and Re-randamisation with Re-initialisation signs applied. We conduct this analysis in the MNIST case, in-line with our original experiments across each layer averaging over 100 models. Reporting the mean and standard error of the mean.
>
>
>
> |      Modification                                         | FC1             | FC2             | FC3             | FC4             | FC5             | FC6             |
> |:------------------------------------------------------|:----------------|:----------------|:----------------|:----------------|:----------------|:----------------|
> | Re-initialisation                                     | 0.2549 +-0.0007 | 0.2151 +-0.0041 | 0.1999 +-0.004  | 0.1945 +-0.0036 | 0.1975 +-0.0033 | 0.1889 +-0.0036 |
> | Re-randomisation                                      | 0.5003 +-0.001  | 0.4998 +-0.0011 | 0.4999 +-0.0013 | 0.4996 +-0.0014 | 0.4995 +-0.0014 | 0.4989 +-0.0023 |
> | Re-randomisation with Re-initialisation signs applied | 0.3238 +-0.0008 | 0.2968 +-0.0029 | 0.2863 +-0.0029 | 0.2825 +-0.0027 | 0.2844 +-0.0025 | 0.2662 +-0.0048 |
>
> Here we see that for the Re-randamisation with Re-initialisation signs applied that the layers activation disagreement reduces significantly from the Re-randamisation to be closer to the Re-initialisation control.
>
> The next table shows the resulting accuracies in these conditions of Re-initialisation, Re-randomisation and Re-randomisation with Re-initialisation signs applied.
>
> | Modification    | FC1               | FC2               | FC3               | FC4               | FC5               | FC6               |
> |:------------------------------------------------------|:------------------|:------------------|:------------------|:------------------|:------------------|:------------------|
> | Re-initialisation                                     | 57.2214 +- 0.9356 | 90.7503 +- 0.3277 | 95.1805 +- 0.1306 | 95.6963 +- 0.0804 | 95.3636 +- 0.1138 | 90.1141 +- 0.3321 |
> | Re-randomisation                                      | 10.1519 +- 0.2175 | 10.3883 +- 0.2726 | 10.5175 +- 0.332  | 9.73 +- 0.4149    | 10.0727 +- 0.4766 | 10.2321 +- 0.6798 |
> | Re-randomisation with Re-initialisation signs applied | 38.7367 +- 0.993  | 83.3944 +- 0.7305 | 89.6716 +- 0.508  | 92.4891 +- 0.3103 | 91.4664 +- 0.3656 | 85.3032 +- 0.5833 |
>
> Once again, we see that the accuracy significantly and substantially increases from the Re-randomisation case, showing that preservation of signs (subnetworks) in line with the lottery ticket hypothesis is a key aspect to differentiating between the robust and critical layer phenomena. We will provide these results and the corresponding Figures in our 4.3 COMPARING TO DATA-BASED METRICS Section.
>
> As a result, we argue that this reaffirms the importance of preserving subnetworks under re-randomisation and re-initialisation and further demonstrates the efficacy of data-based metrics to disambiguate between the subnetwork importance of layers in neural networks.
>
> We thank the reviewer for asking this insightful question, as it has led to improving our understanding of what is preserved with re-initialisation and re-randomisation and better position this work of understanding fundamental phenomena within neural networks.

---

> ### Author Response · Authors · 2025-11-20
> **Response 4**
>
> **Q3. While the call to explore alternative methods is valuable, do the authors have concrete recommendations?**
>
> Thank you for providing the opportunity to clarify our recommendations based on our study's insights. We will add these to the paper such that the takeaways from our work are more concrete and easier to access.
>
> Our recommendations are as follows:
>
> 1. Avoid using data-free metrics as they only possess a spurious relation with model performance and are not a necessary prerequisite of a well-generalising model.
> 2. To improve the utility of data-free metrics, they need to ensure that any new metrics are able to meaningfully capture the performance difference offered by re-initialisation and re-randomisation.
> 3. In the pursuit of understanding neural networks fundamentally, we need to consider the crucial interplay between weights and data, as only when considering them together can studies begin to understand causal factors of generalisation.
> 4. And finally, that existing data-based approaches for interpretability show far more promise as they are capable of not only capturing the difference between re-initialisation and re-randomisation and therefore should be championed over spurious data-free metrics.
>
> ## References:
>
> [1]  Dinh, L., Pascanu, R., Bengio, S. and Bengio, Y., 2017, July. Sharp minima can generalize for deep nets. In International Conference on Machine Learning (pp. 1019-1028). PMLR.
>
> [2] Zhang, C., Bengio, S., Hardt, M., Recht, B. and Vinyals, O., 2016. Understanding deep learning requires rethinking generalization. arXiv preprint arXiv:1611.03530.
>
> [3] Power, A., Burda, Y., Edwards, H., Babuschkin, I. and Misra, V., 2022. Grokking: Generalization beyond overfitting on small algorithmic datasets. arXiv preprint arXiv:2201.02177.
>
> [4] Martin, C.H. and Mahoney, M.W., 2017. Rethinking generalization requires revisiting old ideas: statistical mechanics approaches and complex learning behavior. arXiv preprint arXiv:1710.09553.
>
> [5] Martin, C.H., Peng, T. and Mahoney, M.W., 2021. Predicting trends in the quality of state-of-the-art neural networks without access to training or testing data. Nature Communications, 12(1), p.4122.
>
> [6] Lu, H., Zhou, Y., Liu, S., Wang, Z., Mahoney, M.W. and Yang, Y., 2024. Alphapruning: Using heavy-tailed self regularization theory for improved layer-wise pruning of large language models. Advances in neural information processing systems, 37, pp.9117-915
>
> [7] Qing, P., Gao, C., Zhou, Y., Diao, X., Yang, Y. and Vosoughi, S., 2024. Alphalora: Assigning lora experts based on layer training quality. arXiv preprint arXiv:2410.10054.
>
> [8] He, D., Jaiswal, A., Tu, S., Shen, L., Yuan, G., Liu, S. and Yin, L., 2025. AlphaDecay: Module-wise Weight Decay for Heavy-Tailed Balancing in LLMs. arXiv preprint arXiv:2506.14562.
>
> [9] Prakash, H.K. and Martin, C.H., 2025, June. Grokking and Generalization Collapse: Insights from HTSR theory. In High-dimensional Learning Dynamics 2025.
>
> [10] Petzka, Henning, Michael Kamp, Linara Adilova, Cristian Sminchisescu, and Mario Boley. "Relative flatness and generalization." Advances in neural information processing systems 34 (2021): 18420-18432.
>
> [11] Liang, T., Poggio, T., Rakhlin, A. and Stokes, J., 2019, April. Fisher-rao metric, geometry, and complexity of neural networks. In The 22nd international conference on artificial intelligence and statistics (pp. 888-896). PMLR.

---

### Official Review · Reviewer_VRWM · 2025-11-01

**Soundness:** 2
**Presentation:** 1
**Contribution:** 1
**Rating:** 2
**Confidence:** 4

**Summary:**

This paper investigates whether data-free metrics can reliably distinguish between "robust" and "critical" layers in neural networks. Robust layers can be re-initialized or re-randomized without affecting model accuracy, while critical layers cannot. The authors conduct empirical experiments across multiple scales MNIST, ImageNet, and GPT2 to test various data-free metrics, including alpha (α), spectral norms, and entropy measures. Data-free metrics fail to distinguish robust from critical layers, and they cannot predict performance differences between re-initialization and re-randomization.

**Strengths:**

- Experiments appear thorough and sound, including ImageNet and GPT2

**Weaknesses:**

- It is difficult to understand exactly what problem this paper is addressing. Are there previous works that are claiming that data-free metrics should be able to distinguish robust layers from critical layers? Why would we expect to be able to solve that task without data? Why is that an important task?
- Martin & Mahoney (2021) focused on predicting model test performance using data-free metrics. It is not clear to me what motivates attempting to connect to the idea of critical layers from Zhang et al. (2022).

**Questions:**

- Most modern architectures use residual skip connections. Is the concept of robust and critical layers applicable to networks with residual connections?
- Why is it useful to be able to identify robust and critical layers with a data-free metric? Why would we expect it to be possible to predict without data in the first place?

---

> ### Author Response · Authors · 2025-11-20
> **Response 1**
>
> Thank you for reviewing our paper and allowing us to further explain and highlight the motivations and importance of our work.
>
> **W1. It is difficult to understand exactly what problem this paper is addressing. Are there previous works that are claiming that data-free metrics should be able to distinguish robust layers from critical layers? Why would we expect to be able to solve that task without data? Why is that an important task?**
>
> Identifying whether data-free metrics can disambiguate between critical and robust layers is of high importance. If data-free metrics could distinguish robust from critical layers, it would allow for extremely computationally cheap and effective mechanisms to remove redundancy in neural networks. As we are in a regime where models reach billions of parameters and inference costs are a large bottleneck to deploy such large systems, this capability is of particular importance for compression.
> Moreover, the ability to use data-free metrics for disambiguation would not only reduce redundancy in neural networks, but also focus interpretability efforts, such as mechanistic interpretability [1], on layers responsible for essential computation, and do so efficiently.
>
> The reason why data-free metrics would be assumed to do this is that the fundamental work surrounding our main metric of exploration, Alpha, provided by Martin & Mahoney (2021), was introduced as a metric that was able to correctly identify “good” (well-trained) and “bad” layers (over-/under-fit layers). A “good” (well-trained) layer can be understood as a layer that is critical for the network to achieve its performance. Here, Martin & Mahoney define a good alpha range within 2-6, with layers with an alpha of <2 being underfit and layers of >6 being overfit. Consequently, this work directly argues that it is possible to disambiguate the quality of a layer based on data-free analysis and that the alpha value of a layer is directly related to a network's generalising capabilities. If these capabilities are robust and true of data-free metrics, the impacts of this alpha would be considerable across compression and interpretability subdomains as we have described above.
>
> Building on these foundations, Martin et al. (2021) [2] also show that models with "good" alpha values have improved generalisation. Therefore, it is especially relevant to study these metrics in the context of the critical and robust layer phenomenon, which alters properties via re-randomisation and re-initialisation and impacts generalisation. This allows us to observe whether data-free metrics can indeed disambiguate this condition.
>
> Furthermore, practically data-free metrics have been used to identify layers to compress via pruning [3] and which layers to fine-tune with LORA [4], application of weight decay for specific layers [5], as well as trying to explain the transition of memorisation to generalisation in grokking [6]. Each of these works has been presented at major conferences and therefore represents a sub-community within the field of AI. The core premise [3,4,5,6] is that the metrics can be used to effectively aid and direct the compression mechanism and understand generalisation.
>
>
> Our work robustly provides negative results for data-free metrics: we find that, when applied to critical and robust layers, these metrics do not reliably distinguish between well-generalising models and others. Instead, data-free metric values only demonstrate a spurious relationship with generalisation under current optimisation methods. Therefore, a specific value of a data-free metric should not be assumed to indicate a model's generalisation performance. This finding should be seriously considered in future studies.

---

> ### Author Response · Authors · 2025-11-20
> **Response 2**
>
> **W2. Martin & Mahoney (2021) focused on predicting model test performance using data-free metrics. It is not clear to me what motivates attempting to connect to the idea of critical layers from Zhang et al. (2022).**
>
> Martin & Mahoney (2021) also state that data-free metrics can identify “good” and “bad” layers; therefore, these metrics have been used to identify layers to compress via pruning [3] and which layers to fine-tune with LORA [4] application of weight decay for specific layers [5], and to explain generalisation fundamentally [6].
>
> As such, the robust and critical layer phenomenon is a perfect test bed as it is an operation that is applied layer-wise and can either cause no impact on generalisation (via re-initialisation) or destroy generalisation (via re-randomisation) capabilities. Given that alpha explicitly tries to quantify if a layer will allow for effective generalisation, it should be able to disambiguate between these conditions.
>
> However, our results show concretely across architecture scales (MLP-GPT2) and dataset sizes (MNIST-ImageNet)  that data-free metrics fail entirely in disambiguating either conditions or re-initialisation or re-randomisation.
> Our work does not just connect Martin & Mahoney (2021) and Zhang et al. (2022) but deliberatively leverages the insights provided by Zhang et al. (2022) to fundamentally show the spurious correlations provided by data-free metrics using metrics defined by Martin & Mahoney (2021), which serves as the strongest evidence against data-free metrics use.
>
>
> In response to reviewer FJTL (**Q.2**) we show that our findings leveraging Zhang et al. (2022) provide strong evidence in line with the lottery ticket hypothesis (LTH), providing more concrete evidence for the fact that data-free metrics do not consider crucial subnetworks that enable performance.
>
> **Q1. Most modern architectures use residual skip connections. Is the concept of robust and critical layers applicable to networks with residual connections?**
>
> In the original paper on robust and critical layers, Zhang et al. (2022) explore a range of architectures (as we state on L110-L112 at the top of page three), including MLPs, VGGs, ResNets, Transformers, Vision Transformers and MLPMixers. ResNets, Transformers, and Vision Transformers explicitly have skip connections and therefore this phenomenon is well documented for modern architectures. To see the original findings on skip connections, see Figure 4 from Zhang et al. (2022) on page 7, where they show results for ResNet-18’s, ResNet-50, ResNet-101 and ResNet-502.
>
> We would like to highlight that in this paper, we use MLPs, ResNet50’s and GPT2 models (both ResNet50 and GPT2 have residual connections) to reproduce our findings across architecture sizes and dataset scales, which is a core and fundamental part of our analysis to refute the effectiveness of data-free metrics.

---

> ### Author Response · Authors · 2025-11-20
> **Response 3**
>
> **Q2 Why is it useful to be able to identify robust and critical layers with a data-free metric? Why would we expect it to be possible to predict without data in the first place?**
>
> ***Firstly, we would like to highlight that it is not our work but the work of Martin & Mahoney (2021) and [2,3,4,5,6] that posits that it would be possible for data-free metrics to disambiguate this condition as they argue that data-free metrics are a robust and reliable predictor of generalisation by identifying “good” and “bad” layers which can be understood as crictical and non crictal layers.
> Fundamentally, metrics provided by RMT and other data-free metrics state that they have predictive capacity over model performance, as we have stated in responses to W1 and W2. For example, the alpha metric has an Alpha value range between 2-6 specifies a well-trained layer.***
>
> As a result, if these metrics are truly robust predictors of generalisation and model performance, it would be expected that any reparameterisation or modification to a layer that does not impact the generalisation capability of a model would not impact the range of scores recorded for these metrics.
>
> However, our results leveraging the setup of  Zhang et al of re-randomisation and re-initialisation show that while re-initialisation and re-randomisation lead to significantly different performance, the two conditions are not differentiated by any data-free metric explored. Provided our results, it can be understood that these metrics only possess a spurious relation with model performance and are not a necessary prerequisite of a well-generalising model.
>
> To contextualise the impact of our work, we liken it to the role of seminal literature in the loss landscape geometry by Dhin et al (2017) [7], which showed that Hessian-based sharpness measures were not reparameterisation invariant and, therefore, that flat minima were correlated with generalisation but not a causal aspect of them. The work presented in [7] fundamentally changed the way people consider loss landscapes and has been cited 996 times for its presentation of negative results for flat minima.
>
> We hope that our work can push the field forward in two core ways.
>
> 1. Providing a strong test ground for the development of data-free metrics that are able to disambiguate between robust and critical layers. Similar to how the work of Dhin et al (2017) [7] led to the development of sharpness metrics Relative Flatness [8] and Fisher Rao norm [9]  that are reparameterisation invariant and now considered state of the art sharpness metrics as a result.
>
> 2. Concrete evidence for prioritising data-based metrics, until the appropriate data-free metrics are discovered, such as those that we examine in the paper to improve efforts in interpretability and compression.
>
> ## References
>
> [1] Hou, Y., Li, J., Fei, Y., Stolfo, A., Zhou, W., Zeng, G., Bosselut, A. and Sachan, M., 2023. Towards a mechanistic interpretation of multi-step reasoning capabilities of language models. arXiv preprint arXiv:2310.14491.
>
> [2] Martin, C.H., Peng, T. and Mahoney, M.W. Predicting trends in the quality of state-of-the-art neural networks without access to training or testing data. Nat Commun 12, 4122 (2021).
>
> [3] Lu, H., Zhou, Y., Liu, S., Wang, Z., Mahoney, M.W. and Yang, Y., 2024. Alphapruning: Using heavy-tailed self regularization theory for improved layer-wise pruning of large language models. Advances in neural information processing systems, 37, pp.9117-915
>
> [4] Qing, P., Gao, C., Zhou, Y., Diao, X., Yang, Y. and Vosoughi, S., 2024. Alphalora: Assigning lora experts based on layer training quality. arXiv preprint arXiv:2410.10054.
>
> [5] He, D., Jaiswal, A., Tu, S., Shen, L., Yuan, G., Liu, S. and Yin, L., 2025. AlphaDecay: Module-wise Weight Decay for Heavy-Tailed Balancing in LLMs. arXiv preprint arXiv:2506.14562.
>
> [6] Prakash, H.K. and Martin, C.H., 2025, June. Grokking and Generalization Collapse: Insights from HTSR theory. In High-dimensional Learning Dynamics 2025.
>
> [7] Dinh, L., Pascanu, R., Bengio, S. and Bengio, Y., 2017, July. Sharp minima can generalize for deep nets. In International Conference on Machine Learning (pp. 1019-1028). PMLR.
>
> [8] Petzka, Henning, Michael Kamp, Linara Adilova, Cristian Sminchisescu, and Mario Boley. "Relative flatness and generalization." Advances in neural information processing systems 34 (2021): 18420-18432.
>
> [9] Liang, T., Poggio, T., Rakhlin, A. and Stokes, J., 2019, April. Fisher-rao metric, geometry, and complexity of neural networks. In The 22nd international conference on artificial intelligence and statistics (pp. 888-896). PMLR.

---

### Author Response · Authors · 2025-11-27
**Reviewer prompt**

Thank you to all reviewers for taking the time to evaluate our work. We have enjoyed responding to your questions and addressing your concerns.

Some particularly helpful experimental requests include:

1) Showing that under an equal training budget, we can enforce good and bad alpha values for all layers of the whole network while having equivalent accuracy.

2) Relating data-based metrics efficacy in distinguishing between robust and critical layers to fundamental views of the lottery ticket hypothesis.

3) Showing that a neural network with layers all within the optimal alpha range can have entirely random accuracy.

The results from the above have been extremely useful in providing stronger empirical evidence for the pitfalls and weaknesses of current data-free metrics, which we hope future literature can overcome.

We understand that this period is particularly busy, but we hope it will be possible to discuss our repossess to your reviews to ensure that we have addressed all of your concerns and can discuss where you felt we have not.

We will integrate all of the key feedback into the revised manuscript once we have confirmation on the above, such that our edits reflect the latest feedback and are in line with your requests.

Thank you again for your efforts in the review process.

---

### Author Response · Authors · 2025-12-02
**Paper updates in response to reviews**

We thank the reviewers for there comments and questions which have improved our work.  We have uploaded the revised PDF which has integrated all suggestions provided by the reviews. In the PDF edits are presented in Blue.

Below we outline the changes that have been made in the PDF and how they address the weakness/questions raised by the reviewers:

1. To VRWM and FJTL weakness 1 and 3, we have better contextualised our work within the introduction and added a statement (Line 076-086) a contextualising our work with respect to Din et al., 2017 and indicating explicitly how we hope this work will push the field forward.

2. FJTL weakness 2, we have added a statement in the conclusion (Line 516-519) indicating why the data-free metrics fail, provided our further analysis presented in Appendix Section D, where we show alpha, in particular, is not casually related to generalisation.

3. FJTL question 2 surrounding the connection to the lottery ticket hypothesis, we have added Appendix B, which provides further evidence for the lottery ticket hypothesis and further demonstrates the efficacy of data-based metrics to disambiguate between the subnetwork importance of layers in neural networks due to their consideration of subnetwork importance for representational alignment.

4. FJTL Question 3 (exploring alternative methods is valuable, do the authors have concrete recommendations), we have added a section to the conclusion (Line 525-539) with concrete recommendations of how future literature should consider our findings to explore.

5.  KcoL weakness 2 and 4 and questions 1 and 2, Layer vs Model statistics, we have added two experimental setups in Appendix D to further our results.

The first shows that a model can have a collection of layers within a specified range and achieve equivalent random accuracy at initialisation. Showing that an average good alpha has no relevance to model performance causally.

The second shows how the model can be trained to equivalent accuracy at the end of the training, regardless of the layer-wise and model-wise alpha value. Further highlighting the lack of a good alpha-rage to 1) predict a well-performing layer (as posited in the original work) and 2) be casually important for generalisation.

6. KcoL questions surrounding the use and interpretation of current data-free metrics, we have added additional citations to the use of data-free metrics on Line(093-094), totalling 10 current papers which claim that data-free metrics have strong predictive capabilities over generalisation, which this paper fundamentally and rigorously refutes.

## **Overall**

In line with the fundamental endeavor of science to disentangle correlation from causation,we show that current data-free metrics are not causally related to generalisation.

We have provided robust and critical evidence that shows that data-free metrics cannot disambiguate between the performance differences provided by critical robust layers with re-initialisation and re-randomisation, regardless of their distinctive impacts on generalisation. These results fundamentally undermine the capacity of data-free metrics to predict generalisation as well as well-trained layers. We verify our findings from a small-scale setting on MNIST to ImageNet dataset and GPT2 model scale.

As a result, our work challenges assumptions of the effectiveness of data-free metrics beyond spurious relations to performance; as such, we recommend that considerable effort be applied to develop data-free metrics that overcome the shortcomings we have identified. In lieu of such metrics our work prompts a movement to focus on data-based metrics that can effectively disambiguate this task. Such prioritisation of approaches can aid the understanding of deep neural networks, layer importance, compression, and interpretability, all of which are imperative endeavors, as powerful and increasingly complex neural networks  are deployed in high-stakes real-world environments.

---

### Meta-Review · Area_Chair_XAJr · 2026-01-06

**Summary:**

The authors studied the data-free metrics in terms of evaluating robust vs critical layers, and predict the performance drop from re-initializing or re-randomizing. Broadly speaking, I'm understanding this paper as providing negative results on the claims made earlier by Martin and Mahoney regarding their data-free metrics, and they are not capable of distinguishing the difference between re-init and re-rand weights.

Reviewers overall found the set of experiments impressive and very careful, which I tend to agree.

On the other hand, reviewers broadly criticized:
- The relevance and scope of the paper, in particular the robust/critical layers are outside the scope of data-free metrics
- Experimental design and comparisons
- Lack of alternatives

Overall, I'm somewhat sympathetic to the authors in the latter criticisms, but I would like to expand more on the first point. Specifically, reviewer KcoL argues that the $\alpha$ metric of Martin and Mahoney only predicts whether or not a weight matrix is "well trained", not whether or not it will be robust or critical. This is a subtle distinction, as the reviewer argues that robustness is more about alignment with other weight matrices rather than about its own training progression.

The authors respond by pointing out that the data-free metric by the authors of Martin and Mahoney is marketed as being capable to determining performance based on identifying poorly trained layers. Also the good $\alpha$ range seem to allow for untrained layers, while the bad $\alpha$ range seem allow for well trained layers.

This is very interesting and I want to thank both reviewer KcoL and the authors for engaging in this discussion deeply. Ultimately, here's my interpretation of the discussion
- It seems like Martin and Mahoney is making a strong claim about $\alpha$'s ability to determine poorly trained layers and performance implications, based on the range argument alone
- However, simultaneously it is also possible for misalignment to be the main issue leading to critical layers, not how well trained it is. This does in fact in my opinion explain the difference in performance between re-init and re-rand

I think one way to measure the effect of re-rand while maintaining alignment is to consider the following procedure:
- Take a weight matrix $W$
- Compute the SVD $W = UDV^\top$
- If you want to measure the effect of re-rand, one can randomly sample the singular values (diagonals of $D$) from the Marchenko--Pastur distribution, which would match the singular value spectrum of a Gaussian matrix
- If you want to just remove the effect of training minus alignment, one can whiten all the singular values, i.e. setting them to $1$.
- In both cases, you can measure the performance

It's not perfect because a significant part of training is learning the alignment, but I think if the $\alpha$ metric is based on random matrix theory, it shouldn't be able to tell alignment directions apart anyways. Just throwing a random idea out there, and hope it could spark more thoughts.

Returning to this paper, while I'm sympathetic to the case the authors are trying to make (i.e. Martin and Mahoney making very strong claims about the $\alpha$ metric), I definitely have to agree with reviewer KcoL's point on alignment. Furthermore, it is difficult to overrule all three negative reviews unless I see something incredibly unjustified in the reviews. Therefore, I will still recommend reject.

That being said, I do think the authors have an interesting project in their hands, and should definitely consider addressing the criticisms before resubmitting again somewhere else.

**Reviewer Concerns:**

- The relevance and scope of the paper, in particular the robust/critical layers are outside the scope of data-free metrics
- Experimental design and comparisons (addressed)
- Lack of alternatives (somewhat addressed)

**Reviewer Scores:**

VRWM - 2
FJTL - 4
KcoL - 4

I don't foresee any reviewers raising their score if the discussion were to have continued.

---

### Decision · Program_Chairs · 2026-01-26

Reject